# JMJD3-mediated senescence is required to overcome stress-induced hematopoietic defects

Yuichiro Nakata [1✉], Takeshi Ueda [2], Yasuyuki Sera[3], Miho Koizumi[3], Katsutoshi Imamura[1], Akinori Kanai [4], Ken-ichiro Ikeda [5], Norimasa Yamasaki[6], Akiko Nagamachi[7], Kohei Kobatake[5], Masataka Taguchi [8], Yusuke Sotomaru [9], Tatsuo Ichinohe[10], Zen-ichiro Honda[11], Takuro Nakamura[12], Ichiro Manabe[1], Toshio Suda[13], Keiyo Takubo [14], Osamu Kaminuma[6] & Hiroaki Honda [3✉]

## Abstract

Cellular senescence in stem cells compromises regenerative capacity, promotes chronic inflammation, and is implicated in aging. Hematopoietic stem and progenitor cells (HSPCs) are responsible for producing mature blood cells, however, how cellular senescence influences their function is largely unknown. Here, we show that JMJD3, a histone demethylase, activates cellular senescence by upregulating *p16^Ink4a* in competition with Polycomb group proteins, and reprograms HSPC integrity to overcome hematopoietic defects induced by replicative and oncogenic stresses. *Jmjd3* deficiency does not alter global H3K27me3 levels, indicating that JMJD3 epigenetically regulates specific and limited JMJD3 targets under stress. JMJD3 deficiency also impairs stem cell potential, proper cell cycle regulation, and WNT pathway activation in HSPCs under stress. These impaired phenotypes are rescued through exogenous and retroviral introduction of *p16^Ink4a*. This JMJD3-p16^INK4a axis in hematopoiesis is age-dependent and is distinct from cellular senescence. Treatment with a selective JMJD3 inhibitor attenuates leukemic potential during cellular senescence. Taken together, these results demonstrate that JMJD3-p16^INK4a mediates cellular senescence and plays critical roles in the functional integrity of HSPCs under stress.

**Keywords** Cellular senescence; Histone demethylase; Hematopoietic stem cell; Stress hematopoiesis
**Subject Categories** Autophagy & Cell Death; Chromatin, Transcription & Genomics; Stem Cells & Regenerative Medicine

## Introduction

Hematopoietic stem and progenitor cells (HSPCs) are capable of self-renewal and differentiation for all blood cell lineages (Orkin and Zon, 2008). Maintaining HSPC stem cell potential is required for supplying mature blood cells throughout the lifetime of individual. Regulatory machineries to balance proliferative and non-proliferative status in HSPCs are critical to avoid exhaustion and to sustain stem cell pools (Li, 2011). Stable cell cycle arrest without stimuli, known as quiescence, is mediated by specific cell cycle regulators, such as p27^KIP1 and p57^KIP2, that are well characterized in HSPCs (Zou et al, 2011; Matsumoto et al, 2011). Cellular senescence is considered to be a type of cell cycle arrest in response to various intrinsic and extrinsic stimuli and is associated with attenuation of stem cell capacity, release of inflammatory cytokines, and spread of senescent cells (Huang et al, 2022). On the other hand, cellular senescence also promotes biological benefits including enhanced kidney regenerative capacity (DiRocco et al, 2014) and promoting insulin production in pancreatic beta cells (Helman et al, 2016). These reports suggest that cellular senescence mediates stem cell fate determination though intricate transcriptional reprogramming and may be tissue- or cell-type dependent. As such, cellular senescence may control stem cell activity in hematopoiesis, though the mechanisms remain unclear.

*CDKN2A*, encoding p16^INK4a and p19^ARF, is a key senescence gene. P16^INK4a negatively regulates cell cycle progression by binding to CDK4/CDK6, thereby inhibiting CyclinD1-CDK4/CDK6 complex formation which phosphorylates RB resulting in release of the cell cycle promoter, E2F (Ewen, 1994). The *CDKN2A locus* is epigenetically regulated by Polycomb group proteins, including Polycomb repressive complex (PRC) 1 and 2 (Neff et al, 2012; Mohammad et al, 2017; Biehs et al, 2013; Bruggeman et al, 2007).

[1]Department of Systems Medicine, Graduate School of Medicine, Chiba University, Chiba, Japan. [2]Department of Biochemistry, Faculty of Medicine, Kindai University, Osakasayama, Japan. [3]Field of Human Disease Models, Major in Advanced Life Sciences and Medicine, Institute of Laboratory Animals, Tokyo Women's Medical University, Tokyo, Japan. [4]Laboratory of Systems Genomics, Department of Computational Biology and Medical Sciences, Graduate School of Frontier Sciences, The University of Tokyo, Chiba, Japan. [5]Department of Urology, Institute of Biomedical and Health Sciences, Hiroshima University, Hiroshima, Japan. [6]Department of Disease Model, Research Institute for Radiation Biology and Medicine, Hiroshima University, Hiroshima, Japan. [7]Department of Animal Experimentation, Foundation for Biomedical Research and Innovation at Kobe, Kobe city, Japan. [8]Department of Hematology, Atomic Bomb Disease Institute, Nagasaki University, Nagasaki, Japan. [9]Natural Science Center for Basic Research and Development, Hiroshima University, Hiroshima, Japan. [10]Department of Hematology and Oncology, Research Institute for Radiation Biology and Medicine, Hiroshima University, Hiroshima, Japan. [11]Department of Beauty & Wellness, Professional University of Beauty & Wellness, Yokohama, Japan. [12]Department of Experimental Pathology, Institute of Medical Science, Tokyo Medical University, Tokyo, Japan. [13]Institute of Hematology & Blood Diseases Hospital, Chinese Academy of Medical Sciences & Peking Union Medical College, Bei Jing Shi, China. [14]Department of Cell Fate Biology and Stem Cell Medicine, Tohoku University Graduate School of Medicine, Sendai, Japan. ✉E-mail: nakatay@chiba-u.jp; honda.hiroaki@twmu.ac.jp

PRC2, which consists of catalytic (EZH2) and non-catalytic (EED and SUZ12) subunits, recognizes trimethylated lysine 27 on histone H3 (H3K27me3) and contributes to gene silencing (Simon and Kingston, 2009). Cell fate of HSPCs is controlled by transcription factors in combination with epigenetic regulators, such as PRC2. Previous reports showed that PRC2 plays essential roles in the functional integrity of HSPCs by regulating expression of target genes (Di Carlo et al, 2019; Margueron and Reinberg, 2011; Radulović et al, 2013). Thus, perturbation of PRC2 function, along with compromised H3K27me3, is directly linked to loss of HSPC activity and/or leukemogenesis. In fact, gain- and loss-of-function mutations in PRC2 constituents were identified in various hematopoietic malignancies (Lohr et al, 2012; Nikoloski et al, 2010; Ueda et al, 2012).

Two distinct demethylases for H3K27 were identified, UTX/KDM6A and JMJD3/KDM6B. These enzymes share a JmjC domain that catalyzes histone lysine demethylation (Hong et al, 2007; Klose et al, 2006). UTX functions as a tumor suppressor in various cancers, including hematopoietic malignancies (Mar et al, 2012; van Haaften et al, 2009). In contrast, JMJD3 is highly expressed in hematopoietic disorders (Ntziachristos et al, 2014; Ohguchi et al, 2017; Wei et al, 2013). Accordingly, these two enzymes may have distinct functions that target different genes in adult hematopoiesis. Indeed, a previous report demonstrated that JMJD3 and UTX play contrasting roles in acute lymphoblastic leukemia (Ntziachristos et al, 2014). JMJD3 was discovered as a key regulator of macrophages under inflammatory and differentiation stimuli (De Santa et al, 2007). Subsequent reports demonstrated that JMJD3 regulates inflammatory gene loci and macrophage polarization through its demethylase activity (De Santa et al, 2009; Satoh et al, 2010). In addition, JMJD3 plays crucial roles in the differentiation and maintenance of various types of stem cells such as embryonic stem cells, mesenchymal stem cells, and neural stem cells (Ohtani et al, 2013; Park et al, 2014; Ye et al, 2012), and its ectopic expression accelerates the differentiation of human induced pluripotent stem cells (iPSCs) (Akiyama et al, 2016).

JMJD3 is also recruited to the *CDKN2A locus* and induces cellular senescence to prevent cancer cell proliferation in response to stress (Lin et al, 2012; He and Sharpless, 2017; Agger et al, 2009; Barradas et al, 2009). Given that the JMJD3-p16$^{INK4a}$ axis is essential for cellular senescence, we expect that Polycomb group proteins and JMJD3 competitively fine-tune expression of *p16$^{INK4a}$* via methylation and demethylation activities to control cellular senescence in HSPCs. However, whether the JMJD3-p16$^{INK4a}$ axis induces cellular senescence in hematopoiesis and how it influences HSPC potential are still mostly unknown. In this study, we generated conditional *Jmjd3* knockout mice and demonstrated that JMJD3 epigenetically regulates expression of *p16$^{INK4a}$* under stress as a senescence inducer and is associated with maintenance of HSPC integrity by gene reprogramming and protection against excessive cell cycle entry.

# Results

## Acquired deletion of JMJD3 induces minimal defects on adult hematopoiesis at steady state

First, to investigate the role of JMJD3 in steady state hematopoiesis, we generated mice in which exons 15–17 of the *Jmjd3* gene, which encode

part of the JmjC domain, were flanked by two *loxP* sites (Fig. EV1A,B). pIpC treatment almost completely deleted exons 15–17 derived transcripts and JMJD3 protein in *Jmjd3$^{flox/flox}$;MxCre$^+$* bone marrow (BM) cells (Fig. EV1C), indicating successful stable ablation of the *Jmjd3* gene product in the hematopoietic system (hereafter, pIpC-treated *Jmjd3$^{flox/flox}$;MxCre$^-$* and *Jmjd3$^{flox/flox}$;MxCre$^+$* mice are referred to as *Jmjd3$^{+/+}$* and *Jmjd3$^{Δ/Δ}$* mice, respectively).

Analysis of the peripheral blood (PB) parameters of *Jmjd3$^{+/+}$* and *Jmjd3$^{Δ/Δ}$* mice showed no obvious changes in white blood cell (WBC) counts, hemoglobin (Hgb) concentration, platelet (Plt) number, or differentiation status of WBCs (Fig. EV2A,B). In addition, percentage of the lineage negative (Lin$^-$) population and cell numbers of HSPC subfractions (Table EV1) in the BM were similar between the two groups, except for a slight decrease in CMP and MEP fractions in *Jmjd3$^{Δ/Δ}$* mice (Fig. EV2C,D). No hematological diseases developed in *Jmjd3$^{Δ/Δ}$* mice during the 1.5 year observation period. These results indicate that *Jmjd3* deficiency does not induce obvious changes in steady state hematopoiesis.

## Loss of JMJD3 impairs long-term repopulating activity of HSPCs under BMT-induced stress

Since no apparent changes were observed under steady state hematopoiesis, we next examined the behavior of *Jmjd3$^{Δ/Δ}$* HSPCs under stress. We first performed serial bone marrow transplantation (BMT) experiments to assess the role of JMJD3 under replicative stress (Fig. 1A). Equal numbers of Lin$^-$, Sca-1$^+$ and c-kit$^+$ (LSK) cells from *Jmjd3$^{+/+}$* and *Jmjd3$^{Δ/Δ}$* mice, which contain similar number of HSCs (Fig. EV2D), were transplanted into lethally irradiated syngeneic mice. In the first transplant, PB chimerism, PB differentiation status, and percentage of the Lin$^-$ population in donor derived BM cells were similar in recipients transplanted with *Jmjd3$^{+/+}$* and *Jmjd3$^{Δ/Δ}$* LSK cells, although donor-derived chimerism in various BM subfractions increased at least 1.4 fold in mice transplanted with *Jmjd3$^{Δ/Δ}$* cells (Fig. 1B). In contrast, *Jmjd3$^{Δ/Δ}$* cells of the second transplant exhibited significantly lower PB chimerism without affecting differentiation, significantly lower chimerism in all BM subfractions, and a markedly reduced Lin$^-$ population in donor-derived BM cells (Fig. 1C).

To further investigate the long-term effects of JMJD3 deletion on HSPC activity, donor-derived BM cells from the second transplant were subjected to a third transplant and colony formation assays were conducted from LSK cells. A significant reduction of donor-derived PB chimerism was observed third transplants with *Jmjd3$^{Δ/Δ}$* cells, and *Jmjd3$^{Δ/Δ}$* LSK cells generated significantly fewer colonies (Fig. 1D). These results indicate that loss of JMJD3 transiently increases proliferative activity on HSPCs during the early phase (first BMT) but eventually impairs their long-term repopulating and colony-forming abilities in the late phases (second to third BMT).

## Loss of JMJD3 impairs leukemogenic activity of HSPCs under oncogene-induced stress

We next examined the role of JMJD3 under oncogenic stress. We introduced *MLL-AF9*, a well-known leukemogenic gene that induces acute myeloid leukemia (AML), coupled with *EGFP* into Lin$^-$, c-kit$^+$ (LK) cells, and EGFP$^+$ cells were subjected to colony replating and transplantation assays (Fig. 2A). At the second round

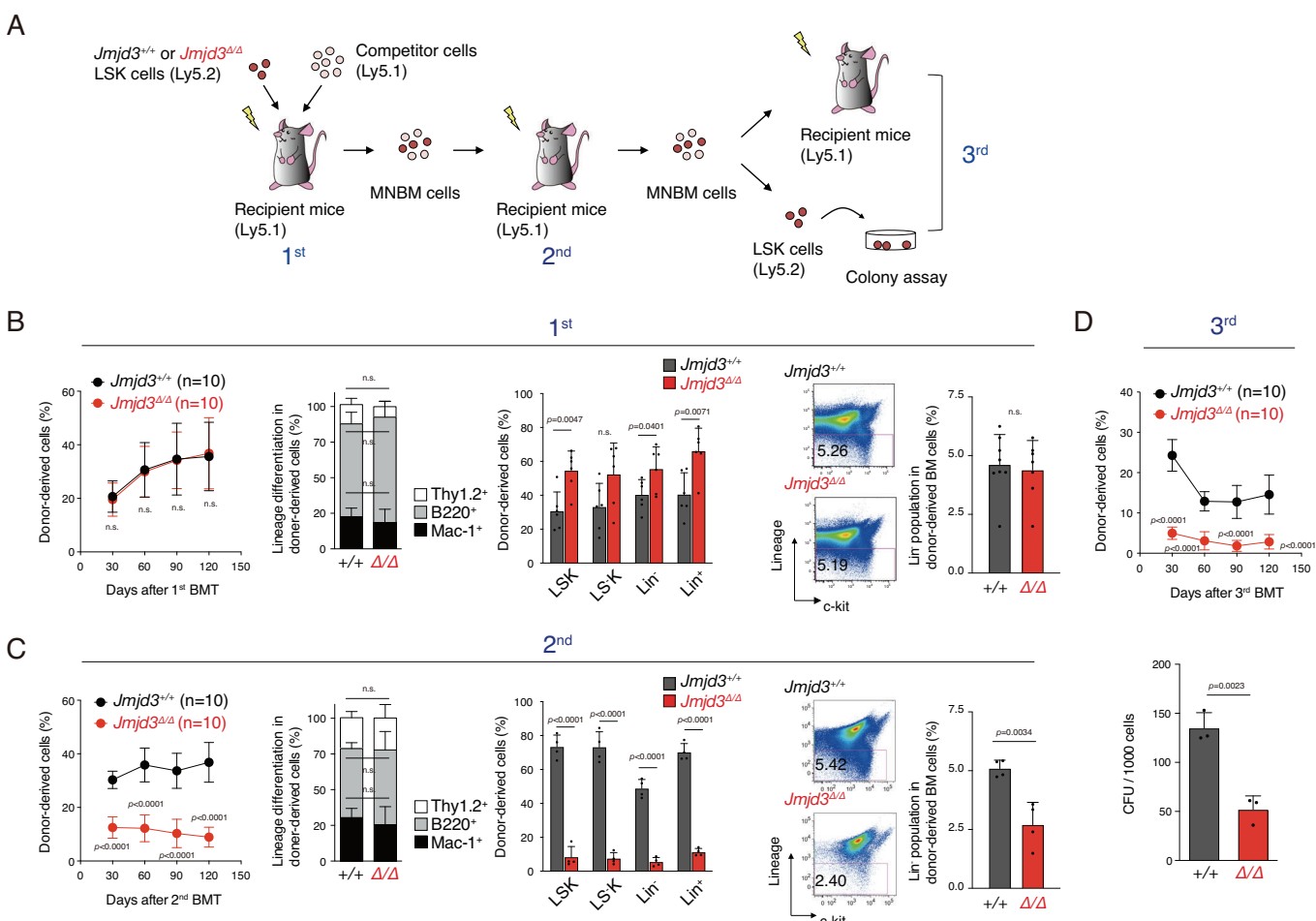

**Figure 1.  Analysis of *Jmjd3*^Δ/Δ HSPCs under replicative stress induced by BMT.**

(A) Schematic diagram of the serial competitive bone marrow (BM) reconstitution experiment. 2.0 × 10³ LSK cells from *Jmjd3*^+/+ and *Jmjd3*^Δ/Δ mice were transplanted into lethally irradiated primary recipients with 2.5 × 10⁵ wild-type competitor mononuclear BM (MNBM) cells. 3.0 × 10⁶ MNBM cells from the first BMT recipients were transplanted in lethally irradiated secondary recipients, and 3.0 × 10⁶ MNBM cells from second recipients were subjected to transplantation into tertiary recipients and colony formation assays. (B) Chimerism and lineage differentiation of donor-derived cells in the peripheral blood (PB) of the first recipients (left and middle left panels, *n* = 10 each) and chimerism of donor-derived cells in BM subfractions including LSK, LS⁻K (Lin⁻, Sca-1⁻, c-kit⁺), Lin⁻, and Lin⁺ of first recipients at 4 months after BMT (middle right panel) (mean ± SD, *n* = 6). Flow cytometric profiles of donor-derived Lin⁻ cells in the BM of the first BMT recipients 4 months after BMT (right panel) (mean + SD, *n* = 5). Student's *t* test was used to calculate *p* value. (C) The same analyses in the second recipients (mean ± SD, *n* = 10). Student's *t* test was used to calculate *p* value. (D) Chimerism of donor-derived cells in the PB of third recipients (mean ± SD, *n* = 10) and colony formation assays of LSK cells from the third recipients at 4 months (mean + SD, *n* = 3). Student's *t* test was used to calculate *p* values. Source data are available online for this figure.

of colony replating, *MA9* expressing (*MA9*) *Jmjd3*^Δ/Δ cells exhibited a 1.7-fold increase in colony formation compared with *MA9* *Jmjd3*^+/+ cells. However, at the third round of replating, *MA9*a *Jmjd3*^Δ/Δ colony formation decreased 3.8-fold (Fig. 2B). In addition, the percentage of LK cells in *MA9* *Jmjd3*^Δ/Δ colonies decreased with replating (Fig. 2C). To evaluate in vivo tumorigenic activity, *MA9* *Jmjd3*^+/+ and *Jmjd3*^Δ/Δ cells were transplanted into lethally irradiated syngeneic mice. Significantly longer survival was observed in recipients transplanted with *MA9* *Jmjd3*^Δ/Δ cells compared with *MA9* *Jmjd3*^+/+ cells (Fig. 2D), and a significantly lower percentage of Lin⁻ cells were observed in the BM of *MA9* *Jmjd3*^Δ/Δ recipients (Fig. 2E).

Leukemia arises from leukemic stem cells (LSCs) and recent studies demonstrated that LSCs transformed by *MLL-AF9* were enriched in the GMP fraction, known as leukemic-GMP (L-GMP)

(Krivtsov et al, 2006). Thus, we performed the same analyses using L-GMPs. Colony numbers of *Jmjd3*^Δ/Δ L-GMPs were significantly lower than those of *Jmjd3*^+/+ L-GMPs (Fig. 2F). In addition, recipients transplanted with *Jmjd3*^Δ/Δ L-GMPs exhibited significantly lower mortality than those from *Jmjd3*^+/+ L-GMPs (Fig. 2G). These findings collectively indicate that JMJD3 deficiency impairs *MLL-AF9*-induced leukemogenesis by perturbing the leukemogenic potential of LSCs.

## JMJD3 epigenetically controls expression of *p16*^Ink4a in HSPCs under stress and competes with PRC2

Since *Cdkn2a* is a known common target of JMJD3 and Polycomb group proteins (Neff et al, 2012; Mohammad et al, 2017; Biehs et al, 2013; Bruggeman et al, 2007; Lin et al, 2012), we rationalized that p16^INK4a

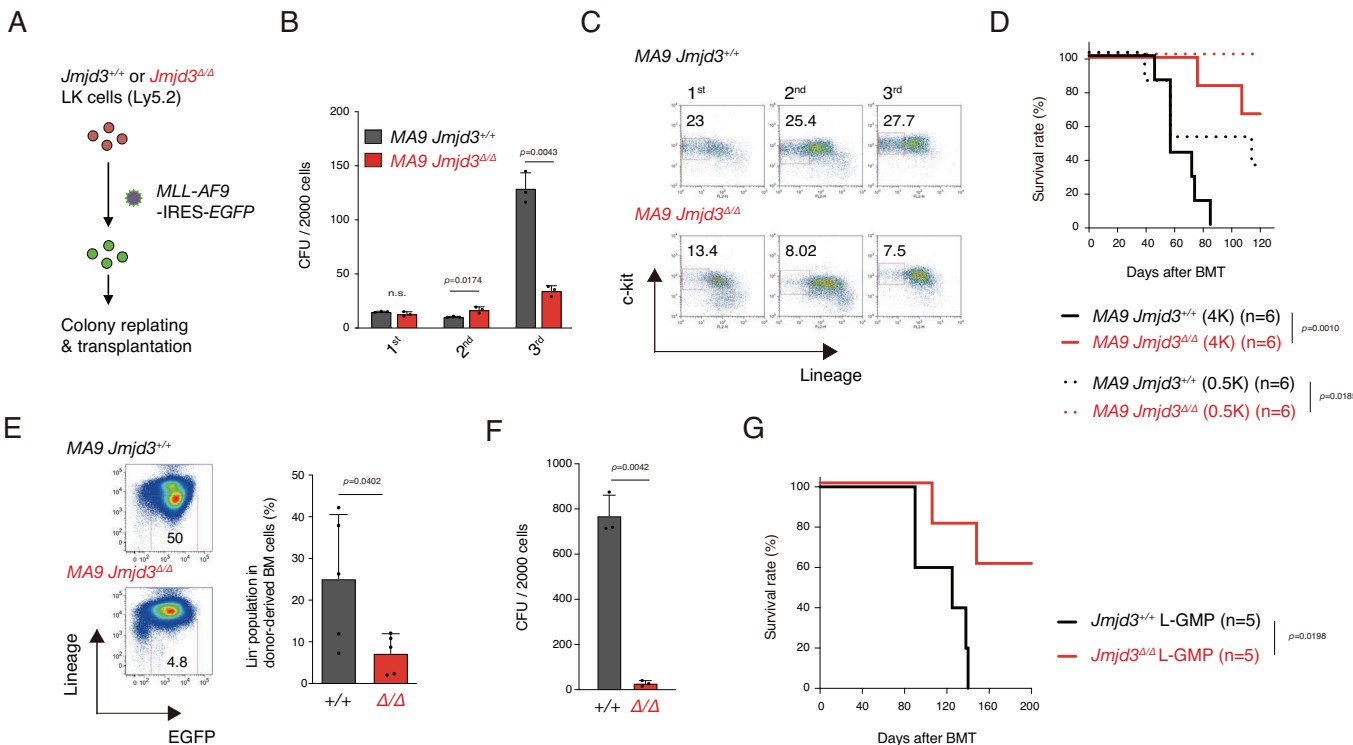

**Figure 2. Analysis of *Jmjd3*<sup>Δ/Δ</sup> HSPCs under oncogenic stress.**

(A) Schematic diagram of *MLL-AF9* oncogene transduction. LK cells from *Jmjd3*$^{+/+}$ and *Jmjd3*$^{Δ/Δ}$ mice were transfected with *MLL-AF9-IRES-EGFP* retrovirus, and EGFP$^+$ cells were subjected to the following assays. (B) Colony replating assays. Bars indicate colony number (CFU; colony-forming unit) in each round of plating (mean + SD, $n = 3$). Student's $t$ test was used to calculate $p$ value. (C) Flow cytometric profiles of colony-forming cells. Cells were stained with c-kit and lineage markers, and the percentages of c-kit$^+$, Lin$^-$ cells in each round are shown. (D) Kaplan–Meier survival curves of mice transplanted with *MA9* expressing (MA9) *Jmjd3*$^{+/+}$ and *Jmjd3*$^{Δ/Δ}$ cells. $4.0 \times 10^3$ (4 K) or $5.0 \times 10^2$ (0.5 K) cells were transplanted into lethally irradiated recipients with $2.5 \times 10^5$ wild-type competitor MNBM cells ($n = 6$). A log-rank test was used to calculate $p$ value. (E) BM cells from mice that developed *MLL-AF9* induced leukemia were stained with EGFP and lineage markers. The percentage of EGFP$^+$, Lin$^-$ cells is shown (mean + SD, $n = 5$). Student's $t$ test was used to calculate $p$ value. (F) Colony replating assay of *Jmjd3*$^{+/+}$ and *Jmjd3*$^{Δ/Δ}$ L-GMPs at the third round of replating (mean + SD, $n = 3$). Student's $t$ test was used to calculate $p$ value. (G) Kaplan–Meier survival curves of mice transplanted with *Jmjd3*$^{+/+}$ and *Jmjd3*$^{Δ/Δ}$ L-GMPs. $2.0 \times 10^2$ L-GMPs were transplanted into lethally irradiated recipients with $2.5 \times 10^5$ wild-type competitor MNBM cells for radioprotection ($n = 5$). A log-rank test was used to calculate $p$ values. Source data are available online for this figure.

played a role in the hematopoietic deficiencies observed under stress caused by loss of JMJD3. To investigate the molecular mechanisms underlying the impaired stem cell potential of *Jmjd3*$^{Δ/Δ}$ HSPCs under stress and their relation with p16$^{INK4a}$, we first examined expression of *CDKI* genes including *Cdkn2a* in three different types of normal hematopoietic cells including: (1) LSK (Steady), LSK cells at steady state, (2) LSK (BMT), LSK cells after second BMT, and (3) L-GMP, cells in the GMP fraction from LK cells after *MLL-AF9* transduction. Drastic upregulation of *Cdkn2a* expression was observed in HSPCs under stress compared with those at steady state, suggesting that *Cdkn2a* has essential roles in hematopoiesis under stress (Fig. 3A). Next, to investigate whether JMJD3 deficiency influences the upregulation of *p16*$^{INK4a}$, we examined *CDKI* expression profiles in *Jmjd3*$^{Δ/Δ}$ cells. We found that expression levels of *p16*$^{Ink4a}$ and *p19*$^{Arf}$, were markedly lower in *Jmjd3*$^{Δ/Δ}$ LSK (BMT) and L-GMP compared with *Jmjd3*$^{+/+}$ but not in *Jmjd3*$^{Δ/Δ}$ LSK (Steady) (Fig. 3B).

Given that JMJD3 upregulates *p16*$^{Ink4a}$ expression through its demethylase activity on H3K27 (He and Sharpless, 2017; Agger et al, 2009; Barradas et al, 2009), we next investigated whether the suppression of *Cdkn2a* in stressed *Jmjd3*$^{Δ/Δ}$ HSPCs is linked to changes in H3K27me3. No obvious changes on global H3K27me3

levels were observed among LSK (Steady), LSK (BMT), and L-GMP from *Jmjd3*$^{+/+}$ and *Jmjd3*$^{Δ/Δ}$ mice by immunostaining (Fig. 3C), prompting us to compare H3K27me3 enrichment in the promoter regions of *Cdkn2a* locus by ChIP-qPCR. No changes were observed in LSK (Steady), but significant enrichment of H3K27me3 was detected in the promoter regions of both *p16*$^{Ink4a}$ and *p19*$^{Arf}$ in *Jmjd3*$^{Δ/Δ}$ LSK (BMT) and L-GMP (Fig. 3D). These data indicate that *Jmjd3* deficiency leads to insufficient demethylation of H3K27me3 at the *Cdkn2a* promoter in HSPCs under stress, which impairs expression of *p16*$^{Ink4a}$ and *p19*$^{Arf}$, despite comparable global H3K27me3 levels. To further investigate the genome-wide distribution of H3K27me3, we performed CUT&RUN sequencing in LSK (BMT) and L-GMP from *Jmjd3*$^{+/+}$ and *Jmjd3*$^{Δ/Δ}$ mice. As expected, no obvious changes were observed on genome-wide H3K27me3 accumulation (Fig. 3E), however, we identified 19,963 and 3,768 unique H3K27me3 peaks in *Jmjd3*$^{Δ/Δ}$ LSK (BMT) and L-GMP, respectively. The genes annotated from these peaks were associated with PRC2 (SUZ12 and EZH2) target genes, indicating that JMJD3 and Polycomb group proteins may have common targetability and competitiveness in hematopoiesis (Fig. 3F,G).

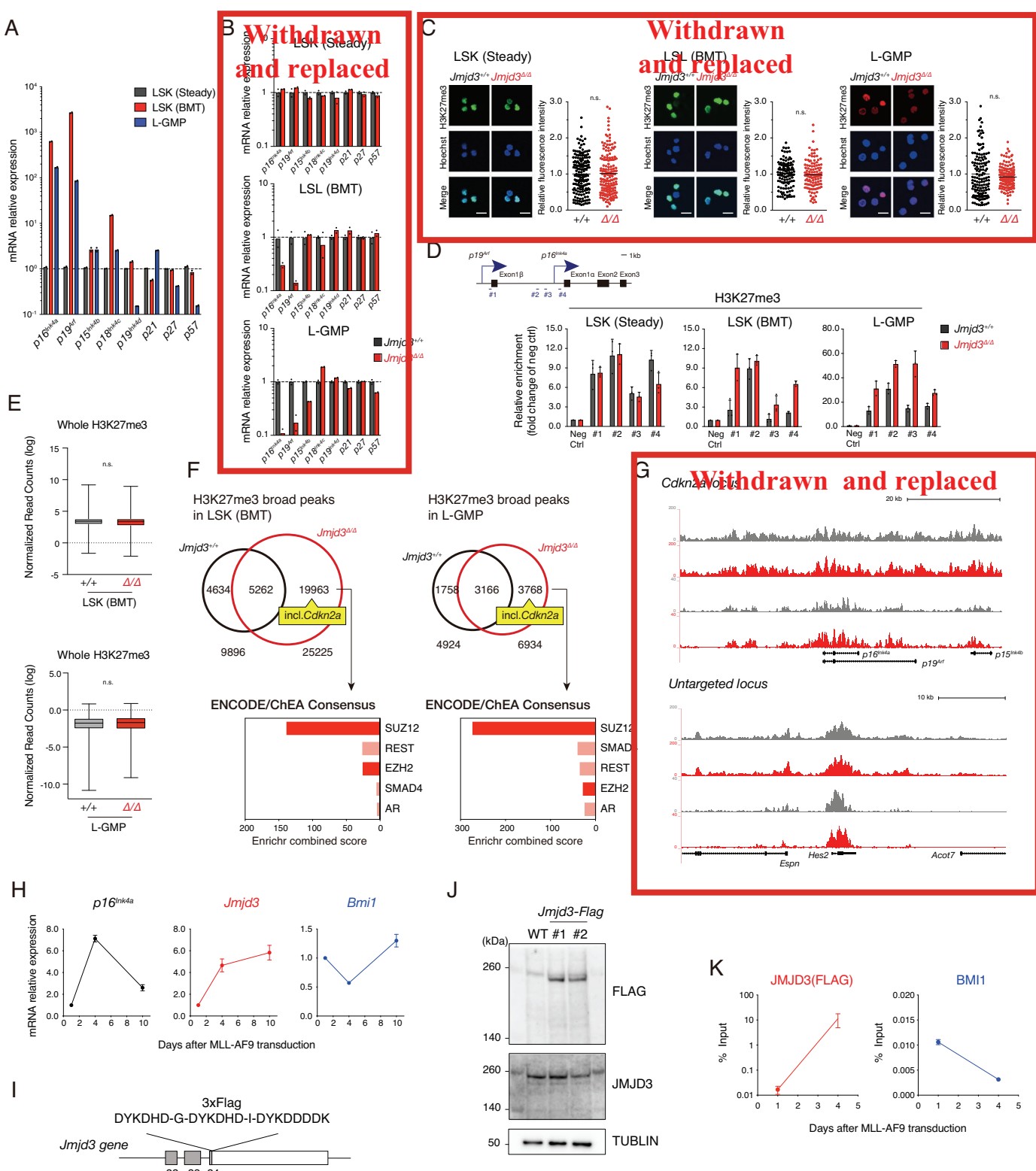

Our results suggest that JMJD3 may not only control expression of *Cdkn2a* genes but may also counteract H3K27me3 at the *Cdkn2a* locus mediated by Polycomb group proteins during stress. To verify this, we examined expression changes of *Jmjd3* (H3K27 demethylase), *Cdkn2a*, and *Bmi1* (Polycomb gene) following *MLL-AF9* transduction. *p16^(Ink4a)* and *Jmjd3* were upregulated whereas *Bmi1* was downregulated at the early phase of oncogenic stress (Fig. 3H). In addition, to investigate the recruitment of JMJD3 and BMI1 to the *Cdkn2a* locus, we generated *Jmjd3-Flag* knock-in (KI) mice in which a 3× Flag tag was inserted into 3′ end of *Jmjd3* immediately upstream of the stop codon (Fig. 3I,J). ChIP-qPCR

**Figure 3.  Demethylase-dependent regulation at the *Cdkn2a* locus in *Jmjd3*+/+ and *Jmjd3*Δ/Δ HPSCs.**

(A) qPCR of CDK inhibitor (CDKI) genes in LSK (Steady), LSK (BMT), and L-GMP. Relative fold-changes to LSK (Steady) are shown on a logarithmic scale (mean + SD, $n = 3$). (B) qPCR of CDKI genes in LSK (Steady), LSK (BMT), and L-GMP of *Jmjd3*+/+ and *Jmjd3*Δ/Δ mice (mean + SD, $n = 3$). (C) Immunofluorescence staining (left panels) and relative fluorescence intensity (right panels) of H3K27me3 in LSK (Steady) ($n = 175$), LSK (BMT) ($n = 117$), and L-GMP ($n = 144$) from *Jmjd3*+/+ and *Jmjd3*Δ/Δ mice. Mean values are indicated as bars. Student's *t* test was used to calculate *p* value. Scale bar, 10 µm. (D) Schematic diagram of the *Cdkn2a* locus. Promoter regions of *p19Arf* and *p16Ink4a* are indicated #1–4. (upper panel). H3K27me3 levels in the promoter regions of *p19Arf* and *p16Ink4a* genes in LSK (Steady), LSK (BMT), and L-GMP of *Jmjd3*+/+ and *Jmjd3*Δ/Δ mice. Results are shown as fold changes relative to the negative control (Neg ctrl), (mean + SD, $n = 3$) (lower panel). (E) Box plot showing the global levels of H3K27me3 in LSK (BMT) and L-GMPs from *Jmjd3*+/+ and *Jmjd3*Δ/Δ mice. Student's *t* test was used to calculate *p* value. (F) Venn diagram showing the distribution of broad H3K27me3 peaks in LSK (BMT) and L-GMPs from *Jmjd3*+/+ and *Jmjd3*Δ/Δ mice (upper panel) and enrichment analysis from unique H3K27me3 peaks *Jmjd3*Δ/Δ L-GMPs with Enrichr on the ENCODE/ChEA database (lower). (G) Genome browser view of H3K27me3 enrichment at *Cdkn2a* and an untargeted locus in LSK (BMT) and L-GMPs from *Jmjd3*+/+ and *Jmjd3*Δ/Δ mice. (H) qPCR analysis of *Jmjd3* (left panel), *p16Ink4a* (middle panel), and *Bmi1* (right panel) in LK cells transduced with *MLL-AF9*. Expression levels are shown relative to day 1 (mean ± SD, $n = 3$). SDs were calculated with technical duplicates. (I) Generation of *Jmjd3-Flag* KI mice. In all, 3× Flag sequences were inserted upstream of the stop codon *Jmjd3*. (J) Immunoblot showing Jmjd3-Flag3 in BM cells of wild-type (WT) and *Jmjd3-Flag* KI mice (#1 and #2). (K) ChIP-qPCR analysis for the enrichment of Jmjd3 and Bmi1 at the indicated *Cdkn2a* promoter regions in LK cells transduced with *MLL-AF9* from *Jmjd3-Flag* cKI mice (mean ± SD, $n = 3$) (see Fig. 3D). Source data are available online for this figure.

analysis revealed that upon *MLL-AF9* transduction, JMJD3 was recruited to the promoter regions of the *Cdkn2a* locus but BMI1 recruitment decreased (Fig. 3K). These data indicate that JMJD3 and Polycomb group proteins control the expression of *Cdkn2a* genes competitively under stress.

## Activation of the JMJD3-p16INK4a axis protects HSPCs from cell cycle over-entry by inducing senescence

To investigate the JMJD3-p16INK4a axis induced cellular phenotype on senescence and cell cycle progression, β-galactosidase staining assay, a marker of cellular senescence (Dimri et al, 1995; Cai et al, 2020), was performed in LSK cells (Steady) and L-GMPs from *Jmjd3*+/+ and *Jmjd3*Δ/Δ in combination with adriamycin (ADR) treatment to enhance senescence induction. Although the population percentage of β-galactosidase+ cells between *Jmjd3*+/+ and *Jmjd3*Δ/Δ LSK cells (Steady) were comparable, the ratio in *Jmjd3*Δ/Δ L-GMPs was significantly reduced (Fig. 4A). Moreover, the percentage of β-galactosidase+ cells in L-GMP is much higher than in LSK (Steady) cells, strongly suggesting that cellular senescence in hematopoiesis may be essential for maintaining stem cell potential in response to stress but not at steady state. Next, we performed BrdU incorporation assays to investigate cell cycle progression. HSPCs under stress with an impaired JMJD3-p16INK4a axis exhibited significantly increased BrdU uptake compared with HSPCs at steady state (Fig. 4B). We observed no correlation between the loss of p16INK4A-mediated cellular senescence and apoptosis in HSPCs under stress (Fig. 4C). Altogether, these data indicate that the cellular senescence induced by activation of the JMJD3-p16INK4a axis is crucial for maintaining stem cell activity in HSPCs under stress by regulating cell cycle entry.

## The JMJD3-p16INK4a axis activates senescence induced reprogramming in HSPCs under stress

To address global gene expression changes induced by JMJD3-p16INK4a axis-mediated senescence between *Jmjd3*+/+ and *Jmjd3*Δ/Δ LSK (Steady), LSK (BMT), and L-GMP, we performed RNA-sequencing (RNA-seq). Genes with more than twofold upregulation or downregulation are shown in Fig. 5A. Except for downregulated genes in LSK (BMT), expression levels in fewer than 2% of all genes were affected by *Jmjd3* deficiency, suggesting that although the defects of *Jmjd3*Δ/Δ HSPCs

could be attributed to specific and limited JMJD3 target genes, *Jmjd3* deficiency under stress greatly influences global gene expression changes compared with *Jmjd3* deficiency at steady state.

We then assessed pathway changes using Gene Set Enrichment Analysis (GSEA). Among KEGG gene sets, we identified both upregulated and downregulated biological pathways (Fig. 5B). Upregulation of SPLICEOSOME and downregulation of CAL-CIUM_SIGNALING_PATHWAY gene sets were commonly observed in *Jmjd3*Δ/Δ LSK (BMT) and L-GMP, suggesting that these pathways are affected by *Jmjd3* deficiency under stress. Notably, both *Jmjd3*Δ/Δ LSK (BMT) and L-GMP showed significant negative enrichment of genes that are upregulated in HSCs and LSCs (EPPERT_CE_HSC_LSC) (Eppert et al, 2011), as well as genes upregulated in quiescent HSCs (GRAHAM_NORMAL_-QUIESCENT_VS_NORMAL_DIVIDING_UP) (Graham et al, 2007). These expression changes were not observed in *Jmjd3*Δ/Δ LSK (Steady) (Fig. 5C). Consistent with Figs. 1, 2, and 4, these findings indicate that *Jmjd3* deficiency impairs stem cell features by perturbing quiescence of HSCs and LSCs and inducing excessive HSPC and LSC proliferation under stress. To further determine whether JMJD3 and Polycomb group proteins competitively regulate their target loci under stress, we investigated possible target gene overlaps between JMJD3 and Polycomb proteins (Bracken et al, 2006; Wiederschain et al, 2007; Douglas et al, 2008; Ben-Porath et al, 2008; Kondo et al, 2008). We found that several gene sets upregulated in Polycomb deficient cells were significantly negatively enriched in *Jmjd3*Δ/Δ LSK (BMT) and L-GMP (Fig. EV3), indicating that this substantial overlap in JMJD3 and Polycomb target genes may enable cells to fine-tune gene expression quickly and reversibly when stress is induced.

As expected, the p16INK4a/RB1 senescent pathway (CHICAS_RB1_-TARGET_SENESCENT) (Chicas et al, 2010) was negatively enriched in *Jmjd3*Δ/Δ LSK (BMT) and L-GMP, but not in *Jmjd3*Δ/Δ LSK (Steady) (Fig. 5D). A major characteristic of senescence associated reprogramming is significant enrichment of canonical WNT signaling (Milanovic et al, 2018). WNT SIGNALING and BETA CATENIN NUC (Schaefer et al, 2008) were negatively enriched in stressed *Jmjd3*Δ/Δ HSPCs (LSK (BMT) and L-GMP) (Fig. 5E). These findings strongly suggest that the JMJD3-p16INK4a axis is a critical regulator of senescence associated reprogramming and contributes to maintaining the stemness of HSPCs under stress.

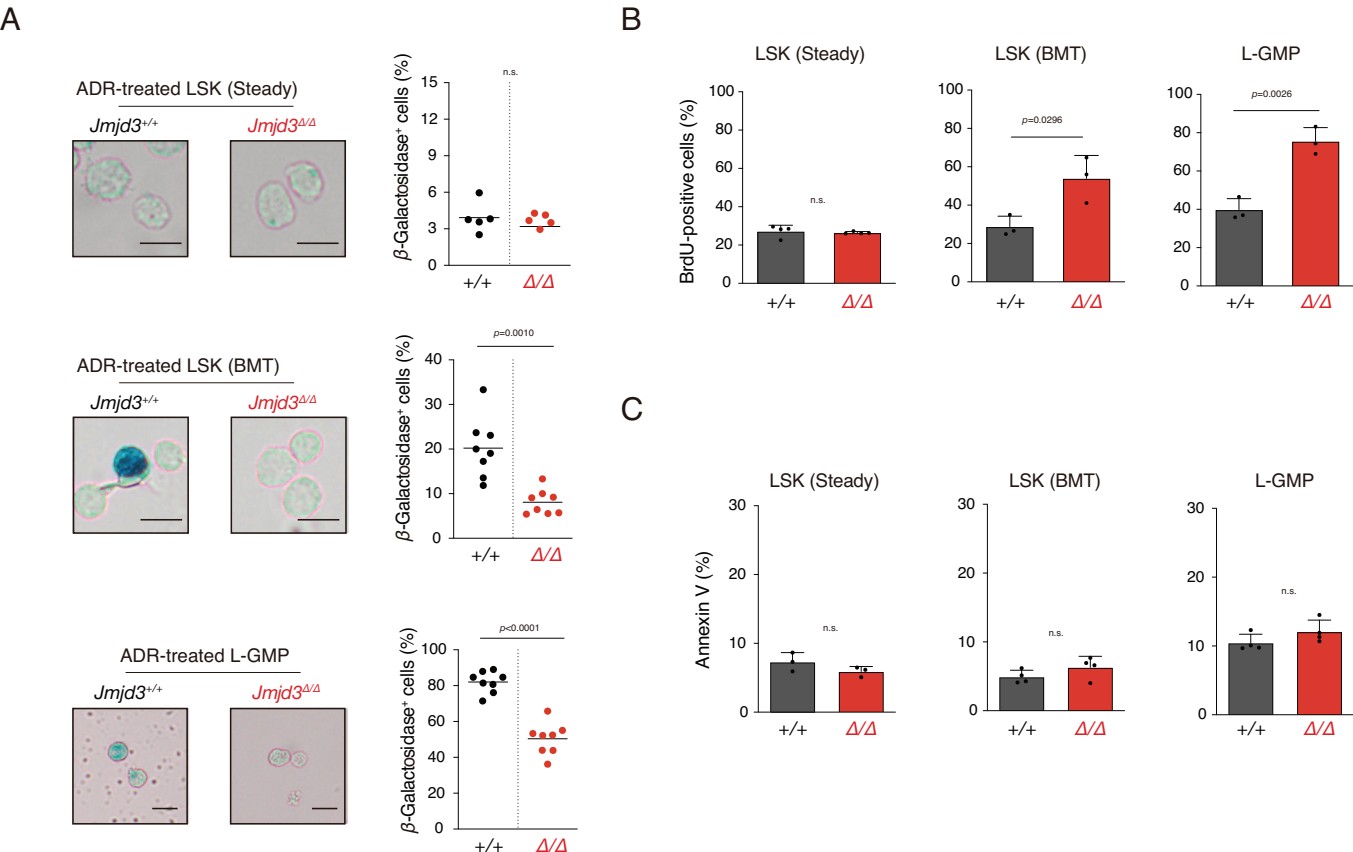

**Figure 4. Jmjd3 deficiency attenuates cellular senescence under stress.**

(A) β-galactosidase staining (left panel) and percentage of β-galactosidase⁺ cells in *Jmjd3⁺/⁺* and *Jmjd3^{Δ/Δ}* LSK (Steady) (*n* = 5), LSK (BMT) (*n* = 8) and L-GMPs (*n* = 8) treated with adriamycin (ADR) (right panel). Mean values are indicated as bars. Student's *t* test was used to calculate *p* value. Scale bar, 10 µm. (B) Flow cytometric analysis of BrdU incorporation in LSK (Steady) (*n* = 4), LSK (BMT) (*n* = 3), and L-GMP (*n* = 3) of *Jmjd3⁺/⁺* and *Jmjd3^{Δ/Δ}* mice (mean + SD). Student's *t* test was used to calculate *p* value. (C) Flow cytometric analysis of Annexin V in LSK (Steady) (*n* = 3), LSK (BMT) (*n* = 4), and L-GMP (*n* = 4) of *Jmjd3⁺/⁺* and *Jmjd3^{Δ/Δ}* mice (mean + SD). Student's *t* test was used to calculate *p* values. Source data are available online for this figure.

## Exogenous p16^{INK4a} rescues impaired repopulating potential of *Jmjd3* deficient HSPCs under replicative stress

We next examined whether exogenous addition of p16^{Ink4a} could recover the defects caused by *Jmjd3* deficiency. We used p16^{INK4a} HIV trans-activating protein (p16-TAT) that enters cells at high efficiency (Fig. 6A) (Krosl et al, 2003). We first confirmed that p16-TAT was detectable in cultured cells for at least 12 h following the addition into the conditioned medium (Fig. 6B) and found that the population of actively cycling cells (BrdU^{high}) *Jmjd3* deficient LSK cells was significantly reduced by the p16-TAT treatment (Fig. 6C). We then allowed *Jmjd3⁺/⁺* or *Jmjd3^{Δ/Δ}* LK cells treated with bovine serum albumin (BSA) or p16-TAT to form colonies and conducted BMT assays (Fig. 6D). Addition of p16-TAT successfully restored the impaired colony forming ability of BSA treated *Jmjd3^{Δ/Δ}* cells (Fig. 6E). Furthermore, a 6.2-fold increase in donor-derived chimerism in the PB was observed in *Jmjd3^{Δ/Δ}* recipients transplanted with p16-TAT compared with BSA treated recipients and no obvious differences were detected in the percentages of

lineage committed cells in the BM of both groups at the end of BMT (Fig. 6F,G). Additionally, at least a 3.5-fold increase in donor-derived chimerism in various BM cell fractions was observed in recipients transplanted with p16-TAT expressing *Jmjd3^{Δ/Δ}* cells compared with BSA treated recipients (Fig. 6H). The Lin⁻ population percentage in p16-TAT treated *Jmjd3^{Δ/Δ}* cells in the BM was comparable to BSA treated *Jmjd3⁺/⁺* cells (Fig. 6I). These results indicate that p16-TAT expression in *Jmjd3^{Δ/Δ}* cells ameliorated the impaired donor-derived chimeras in PB without affecting lineage differentiation of BMT recipients.

Since p16^{INK4a} inhibits CyclinD1-CDK4/CDK6 complex formation (Ewen, 1994), we then examined cell cycle inhibition in *Jmjd3^{Δ/Δ}* cells using a dominant negative form of CyclinD1, CyclinD1^{T156A}, that is catalytically inactive and unable to phosphorylate RB (Diehl and Sherr, 1997). *Empty*, *Ccnd1^{WT}* or *Ccnd1^{T156A}* coupled with *EGFP*, was introduced into *Jmjd3⁺/⁺* or *Jmjd3^{Δ/Δ}* LK cells, and EGFP⁺ cells were transplanted into recipient mice (Fig. EV4A). Intriguingly, although there was less chimerism in *Ccnd1^{WT}* transduced *Jmjd3^{Δ/Δ}* cells than in *empty* transduced *Jmjd3^{Δ/Δ}* cells, *Ccnd1^{T156A}* transduced *Jmjd3^{Δ/Δ}* cells exhibited nearly the same repopulation and differentiation abilities as

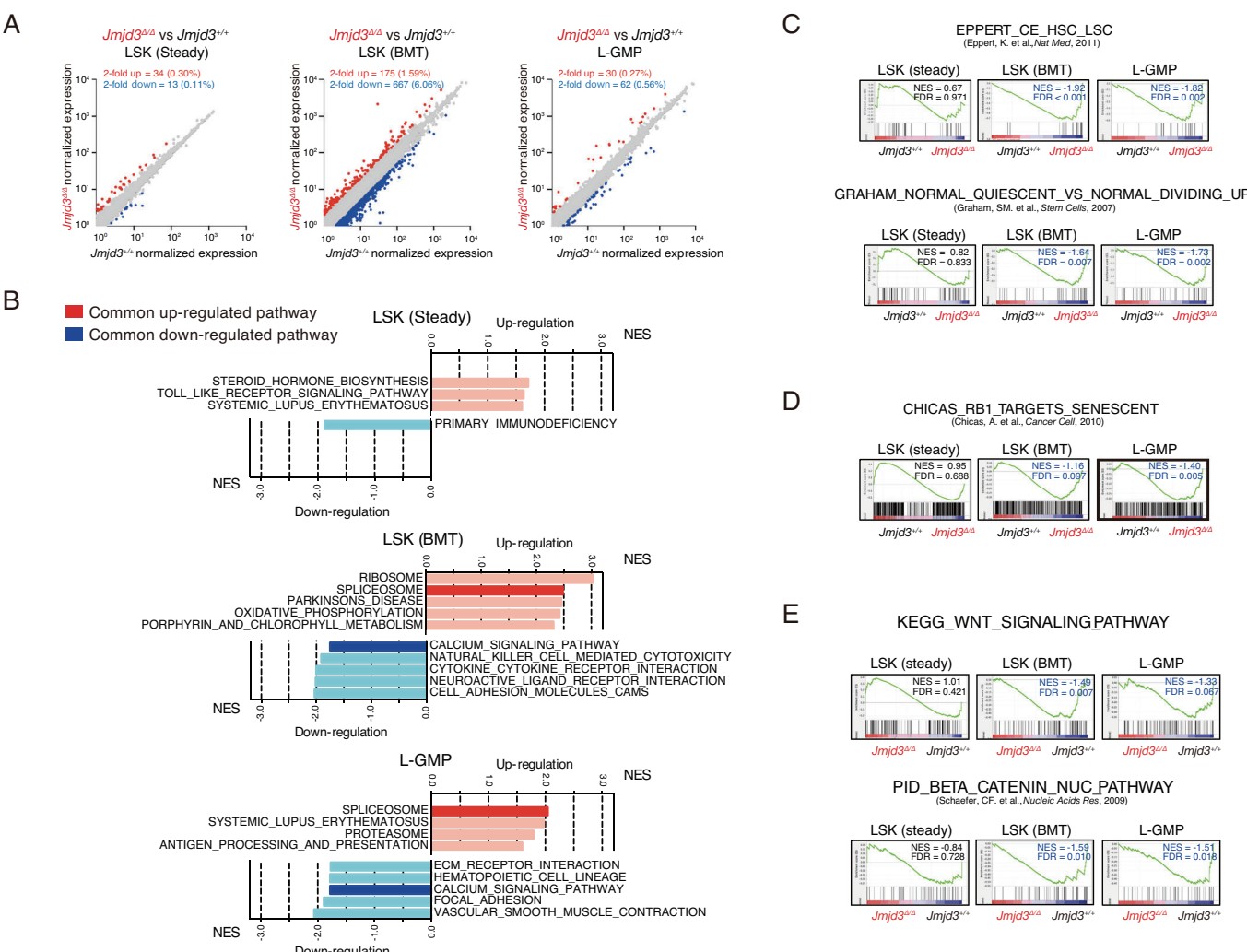

**Figure 5. Expression and pathway analysis of senescence-associated genes.**

(A) Scatter plots comparing normalized expression of individual genes (RPKM > 1) in LSK (Steady), LSK (BMT), and L-GMP of $Jmjd3^{+/+}$ and $Jmjd3^{\Delta/\Delta}$ mice. Numbers and percent of genes upregulated or downregulated more than twofold in $Jmjd3^{\Delta/\Delta}$ cells compared with $Jmjd3^{+/+}$ cells are shown as red and blue dots, respectively. (B) The top five most positively and negatively enriched KEGG pathways in LSK (Steady) (upper), LSK (BMT) (middle), and L-GMP (lower) from $Jmjd3^{\Delta/\Delta}$ mice compared with $Jmjd3^{+/+}$ mice (FDR < 0.25). Common upregulated and downregulated pathways between LSK (BMT) and L-GMP are shown as red and blue bars, respectively. (C) GSEA plots of LSK (Steady), LSK (BMT), and L-GMP in the indicated gene sets (top, genes commonly upregulated in human HSCs and LSCs; bottom, genes commonly upregulated in quiescent human CD34[+] hematopoietic cells). Enrichment in $Jmjd3^{\Delta/\Delta}$ cells relative to $Jmjd3^{+/+}$ are shown with NES and FDR values. (D) GSEA plots of LSK (Steady), LSK (BMT), and L-GMP in genes commonly upregulated through a p16[INK4a]/RB1 pathway. Results are shown with NES and FDR values. (E) GSEA plots of WNT related pathways (KEGG_WNT_SIGNALING_PATHWAY (upper) and PID_BETA_CATENIN_NUC_PATHWAY (lower) in LSK (Steady), LSK (BMT), and L-GMP. $Jmjd3^{\Delta/\Delta}$ cells compared with $Jmjd3^{+/+}$ are shown with NES and FDR values.

empty transduced $Jmjd3^{+/+}$ cells (Fig. EV4B,C). This finding was observed despite incomplete rescue of donor-derived percentages in BM subfractions (Fig. EV4D). These results collectively indicate that introduction of cellular senescence by exogenous addition of p16[INK4a] or retroviral transduction of an inactive form of CyclinD1 is capable of restoring and maintaining the stem cell potential of $Jmjd3^{\Delta/\Delta}$ HSPCs under replicative stress.

We next attempted to rescue impaired LSC potential due to $Jmjd3$ deficiency using retroviral transduction of $p16^{Ink4a}$. $Jmjd3^{+/+}$ and $Jmjd3^{\Delta/\Delta}$ LK cells were transduced with $MLL$-$AF9$ coupled with $EGFP$. EGFP[+] cells were transduced with empty or $p16^{Ink4a}$ coupled with

Kusabira Orange (KO), and double-positive cells were subjected to colony replating and BMT assays (Fig. 6J). The reduced Lin[−] population in MA9 $Jmjd3^{\Delta/\Delta}$ cells was successfully rescued by $p16^{Ink4a}$ transduction (Fig. 6K). In addition, colonies of $Jmjd3^{\Delta/\Delta}$ L-GMPs were restored by the introduction of $p16^{Ink4a}$ (Fig. 6L). Moreover, $p16^{Ink4a}$ expressing $Jmjd3^{\Delta/\Delta}$ L-GMPs exhibited comparable in vivo leukemogenic potential relative to $Jmjd3^{+/+}$ L-GMPs, as evidenced by similarly shortened survival rates of recipient mice (Fig. 6M). Taken together, retroviral transduction of $p16^{Ink4a}$ successfully rescued the impaired leukemic potential of $MLL$-$AF9$-transformed $Jmjd3^{\Delta/\Delta}$ HSPCs via induction of cellular senescence.

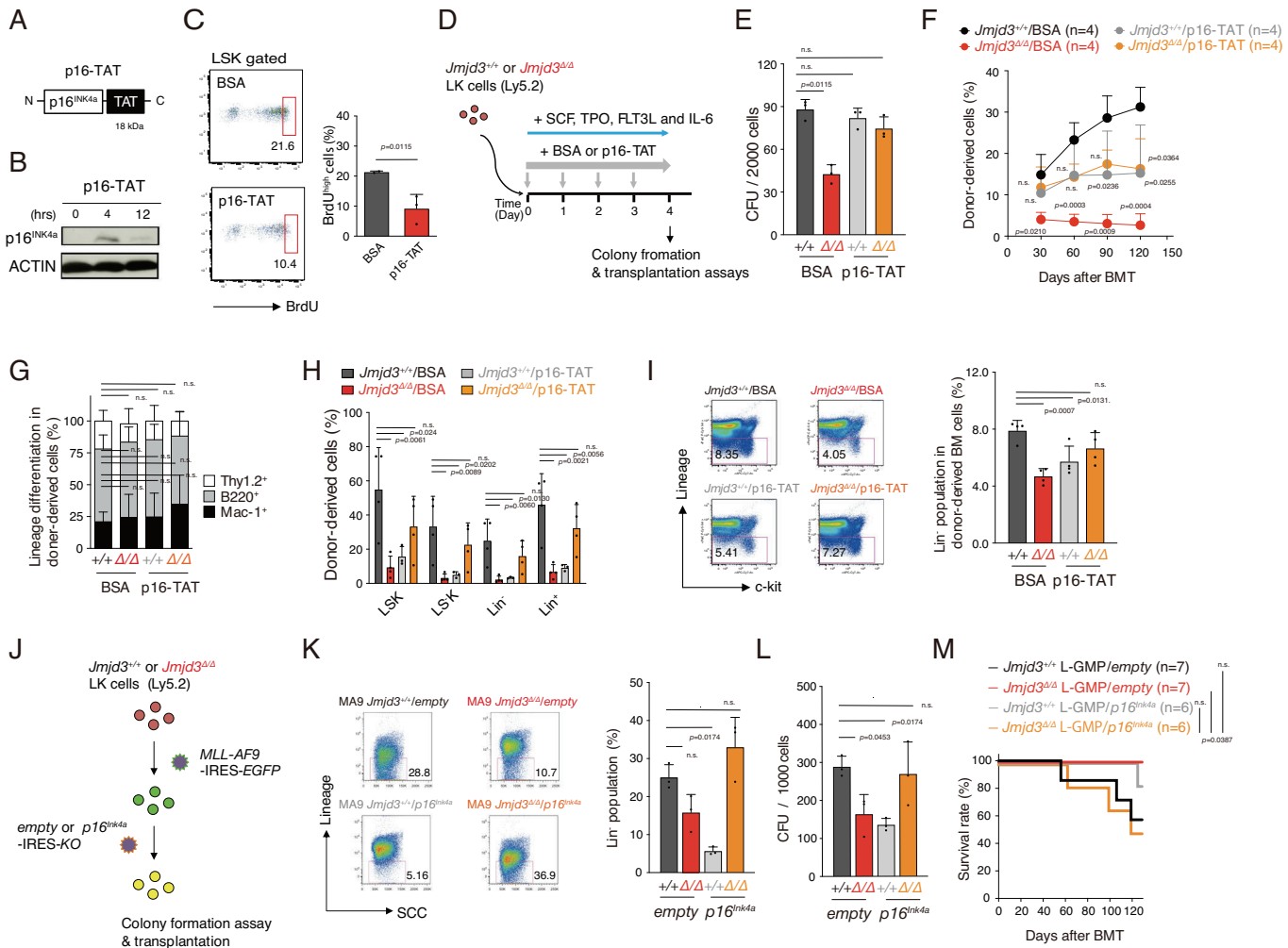

**Figure 6. Rescue of stress-induced defects in *Jmjd3*^Δ/Δ HSPCs by exogenous and retroviral expression of *p16*^Ink4a.**

(A) Schematic diagram of the p16^INK4a-TAT fusion protein (p16-TAT, left panel). (B) Immunoblot showing time-dependent stability of p16-TAT (50 nM) in cultured LK cells. (C) Flow cytometric profiles and percentages of BrdU incorporation in cultured LSK cells treated with BSA or p16-TAT (50 nM) (mean + SD, n = 3). Student's *t* test was used to calculate *p* value. (D) Schematic of experimental procedure. *Jmjd3*^+/+ and *Jmjd3*^Δ/Δ LK cells treated with BSA or p16-TAT (50 nM) in a cytokine cocktail were subjected to colony forming and BMT assays. (E) Colony formation assay of *Jmjd3*^+/+ and *Jmjd3*^Δ/Δ LK cells treated with BSA or p16-TAT for 4 days (mean + SD, n = 3). Dunnett's test was used to calculate *p* value. (F) Donor-derived chimerism in the PB of recipient mice. 5.0 × 10^4 *Jmjd3*^+/+ or *Jmjd3*^Δ/Δ LK cells treated with BSA or p16-TAT for 4 days were transplanted into lethally irradiated recipients with 2.5 × 10^5 wild-type competitor MNBM cells for radioprotection (mean + SD, n = 4). Dunnett's test was used to calculate *p* value. (G) Percent of donor-derived, lineage committed (Thy1.2^+, B220^+, or Mac-1^+) cells in the PB of recipient mice 4 months after BMT (mean + SD, n = 4). Dunnett's test was used to calculate *p* value. (H) Percent of donor-derived LSK, LS^−K, Lin^−, and Lin^+ cells in the BM of recipient mice at 4 months after BMT (mean + SD, n = 4). (I) Flow cytometric profiles and percentages of Lin^− cells in the donor-derived BM of recipient mice 4 months after BMT (mean + SD, n = 4). Dunnett's test was used to calculate *p* value. (J) Schematic diagram of *MLL-AF9* and *p16*^Ink4a co-transduction. LK cells from *Jmjd3*^+/+ and *Jmjd3*^Δ/Δ mice were transfected with the *MLL-AF9-IRES-EGFP* retrovirus. EGFP^+ cells were further transfected with *empty*- or *p16*^Ink4a-IRES-KO (Kusabira Orange) retrovirus, and double-positive cells were subjected to the following assays. (K) Flow cytometric profiles of Lin^− cells in *MLL-AF9 Jmjd3*^+/+ and *Jmjd3*^Δ/Δ leukemic cells transfected with *empty* or *p16*^Ink4a (mean + SD, n = 3). Dunnett's test was used to calculate *p* value. (L) Colony forming assay of *Jmjd3*^+/+ and *Jmjd3*^Δ/Δ L-GMPs transfected with *empty* or *p16*^Ink4a. Bars indicate the colony numbers at the third round of replating (mean + SD, n = 3). Dunnett's test was used to calculate *p* value. (M) Kaplan–Meier survival curves of mice transplanted with *Jmjd3*^+/+ and *Jmjd3*^Δ/Δ L-GMPs transfected with *empty* or *p16*^Ink4a. 2.0 × 10^2 L-GMPs were transplanted into lethally irradiated recipients with 2.5 × 10^5 wild-type competitor MNBM cells (n = 6–7). A log-rank test was used to calculate *p* values. Source data are available online for this figure.

## Roles of JMJD3 in aging related accumulation of *p16*^Ink4a

Increasing evidence links senescence and aging to epigenetic alterations (Sen et al, 2016), and p16^INK4a is reported to be closely associated with stem cell aging (Janzen et al, 2006; Krishnamurthy et al, 2006; Molofsky et al, 2006). We first examined the expression of *p16*^Ink4a in young and aged *Jmjd3*^+/+ and *Jmjd3*^Δ/Δ HSPCs

(Fig. 7A). We observed aging associated upregulation of *p16*^Ink4a in *Jmjd3*^+/+ HSPCs (Fig. 7B), as previously reported (Janzen et al, 2006). Interestingly, no such effects were detected in *Jmjd3*^Δ/Δ cells. In addition, significant enrichment of H3K27me3 was detected at the *p16*^Ink4a promoter in aged *Jmjd3*^Δ/Δ HSPCs compared with aged *Jmjd3*^+/+ cells (Fig. 7C). We then performed competitive repopulation assays using LT-HSCs of young (2 M), mature (18 M) and aged

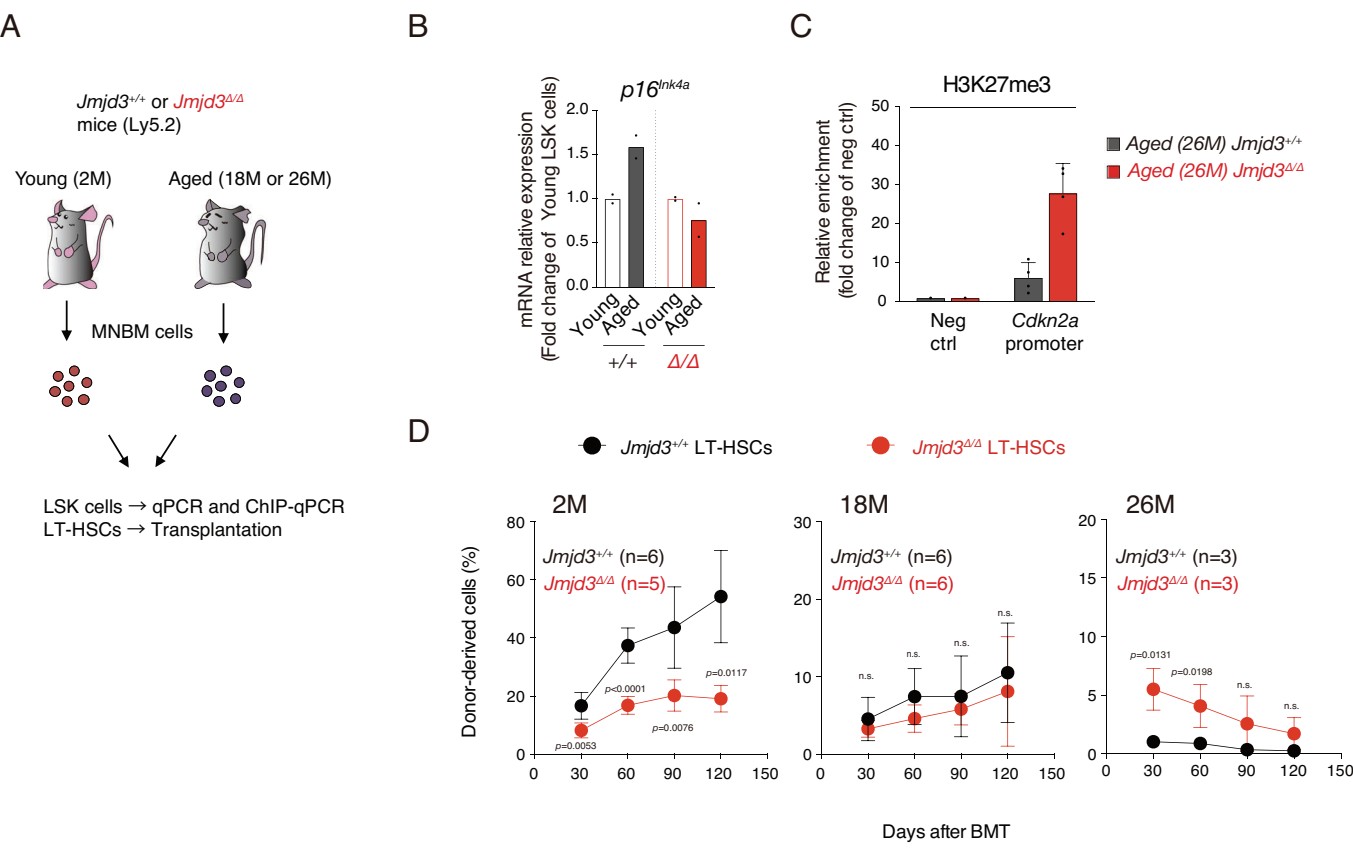

**Figure 7. JMJD3 contributes to cellular aging by regulating *p16^Ink4a*.**

(A) Schematic diagram of the experiments with young and aged LSK cells and LT-HSCs. (B) qPCR analysis of *p16^Ink4a* in LSK cells of young (2 M) and aged (26 M) *Jmjd3^+/+* and *Jmjd3^Δ/Δ* mice (mean + SD, *n* = 3). (C) H3K27me3 enrichment in the promoter region of *Cdkn2a* (see Fig. 3D) in LSK cells from aged (26 M) *Jmjd3^+/+* and *Jmjd3^Δ/Δ* mice. Results are shown as fold changes relative to a negative control (Neg ctrl) (mean + SD, *n* = 3). (D) Competitive repopulation assays. 5.0 × 10² LT-HSCs from young (2 M) (*n* = 6–7), mature (18 M) (*n* = 6–7), and aged (26 M) (*n* = 4) *Jmjd3^+/+* and *Jmjd3^Δ/Δ* mice were transplanted into lethally irradiated recipients with 2.5 × 10⁵ competitor MNBM cells (mean ± SD). Student's *t* test was used to calculate *p* values. Source data are available online for this figure.

(26 M) *Jmjd3^+/+* and *Jmjd3^Δ/Δ* mice. When compared with *Jmjd3^+/+* cells, the PB chimerism of *Jmjd3^Δ/Δ* LT-HSCs was significantly lower at 2 M, comparable at 18 M, and increased 5.5-fold in the early phase following BMT at 26 M (Fig. 7D), closely resembling the results reported in *p16^Ink4a* deficient HSCs (Janzen et al, 2006). These findings indicate that *Jmjd3* contributes not only to cellular senescence mediated by the JMJD3-p16^INK4a axis, but also to aging associated accumulation of *p16^Ink4a* in a demethylase-dependent manner.

## GSK-J4, a JMJD3 inhibitor, suppresses LSC potential by inhibiting the JMJD3-p16^INK4a axis

Given that almost no changes were detected in steady-state hematopoiesis by JMJD3 deficiency, we reasoned that functional inhibition of JMJD3 during cellular senescence may be effective against leukemogenesis and lead to attenuation of LSC potential without major effects on normal hematopoiesis. Thus, we hypothesized that GSK-J4, a JMJD3 demethylase inhibitor (Ntziachristos et al, 2014; Hashizume et al, 2014; Lochmann et al, 2018), is a potential treatment option for leukemia.

To this end, we transduced *MLL-AF9* into wild-type LK cells and treated with GSK-J4 at the early phase of leukemogenesis (Fig. EV5A). *MA9* cells treated with 10 μM of GSK-J4 exhibited a significantly reduced stem cell population (Fig. EV5B). We next investigated transcriptional changes in L-GMPs following treatment with GSK-J4. Consistent with *Jmjd3^Δ/Δ* L-GMPs, we observed reduced expression *Cdkn2a* in GSK-J4 treated L-GMPs (Fig. EV5C). Although no significant difference of global H3K27me3 levels was detected, we observed enrichment of H3K27me3 at the *p16^Ink4a* promoter in GSK-J4 treated L-GMPs (Fig. EV5D,E). Similarly, we identified 24 commonly downregulated pathways between GSK-J4 treated and *Jmjd3^Δ/Δ* L-GMPs (Fig. EV5F and Table EV2). The previously identified representative gene sets significantly suppressed by *Jmjd3* deficiency in stressed HSPCs (Fig. 5) were also negatively enriched in GSK-J4 treated L-GMPs (Fig. EV5G). In addition, the population of BrdU⁺ cycling cells exhibited a 1.3 fold increase in GSK-J4 treated L-GMPs compared with DMSO treated L-GMPs (Fig. EV5H). Altogether, these data indicate that chemical inhibition of JMJD3 suppresses cellular senescence via impaired activation of *Cdkn2a* genes, consistent with features observed in *Jmjd3^Δ/Δ* L-GMPs.

Finally, we tested the effects of GSK-J4 on development of AML in vivo. We performed Kaplan–Meier survival analysis, which revealed a statistically significant difference in the survival periods between the groups. The onset of AML was significantly delayed in recipient mice transplanted with *MA9* cells treated with GSK-J4 in vitro compared with those treated with DMSO (Fig. EV5I). Next, recipient mice transplanted with *MA9* cells were treated in vivo with DMSO or GSK-J4 by intraperitoneal injection at the early phase of leukemogenesis and observed increased survival of GSK-J4 treated mice (Fig. EV5J). This data indicates that selective inhibition of JMJD3 during cellular senescence can potentially suppress leukemic stem cell activity.

# Discussion

Cellular senescence is an intricate biological process with bilateral characteristics (Huang et al, 2022). The precise mechanisms behind cellular senescence in HSPCs and whether it positively or negatively controls stem cell functions are largely unknown. In this study, we highlighted the benefits of JMJD3-mediated cellular senescence in HSPCs, such as enhancing HSPC functional integrity to overcome stress-induced hematopoietic defects through reduction of H3K27me3 at the *Cdkn2a* locus in a demethylase dependent manner, thereby upregulating the expression of *p16^{Ink4a}*. JMJD3-mediated cellular senescence is considered a temporal and reversible cell cycle arrest. Recent works also point to cellular senescence as a critical step in promoting stem or oncogenic properties via a reversible cell cycle arrest and gene reprogramming under cellular senescence (Milanovic et al, 2018; Guccini et al, 2021).

The roles of the *Cdkn2a* in HSCs and LSCs remain controversial. Studies demonstrated that derepressed expression of *Cdkn2a* exerts deleterious effects on stem cell activity. HSCs and LSCs deficient in epigenetic genes, such as *Bmi1* and *Moz*, exhibit impaired self-renewal activity due to derepression of *Cdkn2a*, and this defect is rescued by genetic ablation of *Cdkn2a* (Oguro et al, 2006; Perez-Campo et al, 2013). In addition, overexpression of CyclinD1-CDK4, the inhibitory targets of p16^{INK4a}, confers a competitive advantage to HSCs (Mende et al, 2015). In contrast, other reports demonstrated that p16^{INK4a} induced cell quiescence is essential for maintaining stem cell activity. A study showed that treatment of HSCs with a CDK4/6 inhibitor accelerates hematologic recovery by protecting HSCs from stress-induced proliferative exhaustion (He et al, 2017). Another report demonstrated that *p16^{INK4a}* expression is required for survival in human papillomavirus associated tumor cells (McLaughlin-Drubin et al, 2013). Our study indicates that exogenous addition of p16^{INK4A} successfully restores impaired stem cell activity in *Jmjd3^{Δ/Δ}* HSCs and LSCs but reduces stem cell potential in *Jmjd3^{+/+}* HSCs and LSCs (Fig. 6). Since *Jmjd3^{+/+}* cells retain stem cell activity via cellular senescence, we postulate that excessive expression of p16^{INK4A} may confer detrimental effects on stem cell potential under stress.

During somatic reprogramming by the Yamanaka factors, p16^{INK4a} expression is also induced but removal of p16^{INK4a} high cells increases reprogramming efficiency (Grigorash et al, 2023), thus quantitative and temporal p16^{INK4a} expression must be precisely regulated for its impact on stem cell integrity. In our study, during the early phase following BMT, including the first

BMT and second replating after *MLL-AF9* transduction, *Jmjd3^{Δ/Δ}* HSPCs with diminished *Cdkn2a* upregulation demonstrated greater reconstitution and proliferative abilities (Figs. 1B and 2B). We also found that stressed induced myelosuppression via treatment with 5-fluorouracil (5-FU) (Takubo et al, 2023), conferred a proliferative advantage to *Jmjd3* deficient HSPCs (Fig. EV6A–C). In contrast, in late phases, including the second BMT and third replating after *MLL-AF9* transduction, *Jmjd3^{Δ/Δ}* cells exhibited significantly reduced reconstitution and proliferative capacity (Figs. 1C and 2B). These results collectively indicate that impaired derepression of *Cdkn2a* confers an initial growth advantage to HSCs and LSCs via accelerated cell cycle progression, but eventually impairs stem properties through over-proliferation, and finally results in functional depletion. Our findings demonstrate that the JMJD3-p16^{INK4a} axis plays essential roles in preventing excessive cell cycle progression and consequential exhaustion of HSCs under stress.

p16^{INK4a} involvement in stem cell aging is well studied. Aging-associated *p16^{INK4a}* accumulation is observed in various types of tissue stem cells, including HSCs (Janzen et al, 2006; Krishnamurthy et al, 2006; Molofsky et al, 2006) (see also Fig. 7), though the underlying mechanisms are not yet comprehensively identified. We found that exogenous stresses upregulate *p16^{Ink4a}* through JMJD3-mediated H3K27 demethylation. This observation suggests the possibility that incomplete p16^{INK4a} downregulation following stress, due to loss of PRC2 function, for instance, and repeated stresses including replicative stress and DNA damage gradually leads to p16 accumulation with age. This idea is supported by our finding that aged *Jmjd3^{Δ/Δ}* HSCs do not exhibit accumulation of *p16^{Ink4a}*, associated with increased H3K27me3 at the *p16^{Ink4a}* promoter. Since *Jmjd3^{Δ/Δ}* LT-HSCs behave similarly to *p16^{Ink4a}* deficient cells in BMT assays, we hypothesize that JMJD3 functions as an upstream regulator of p16^{INK4a} in HSC aging.

Acute leukemia is relapsed in a majority of patients after chemotherapy and the relapse emerges from an immature, drug resistant population of cells, LSCs. The ultimate goal of leukemia treatment is to eradicate LSCs. By using an *MLL-AF9* induced leukemia model, we demonstrate that selective inhibition of JMJD3 under cellular senescence is promising for suppressing the maintenance and proliferative ability of LSCs through derepression of *p16^{Ink4a}* in a demethylase activity dependent manner (Fig. EV5). Indeed, AML cells treated with chemotherapy are in an induced senescent-like phenotype and post-senescent cells give rise to relapsed AML with enhanced LSC activity (Duy et al, 2021). Thus, treatment with a JMJD3 inhibitor such as GSK-J4 alongside chemotherapy during cellular senescence in leukemia cells may offer a promising therapeutic option. This concept also suggests that LSC activity is not fixed but rather flexible and modulated by epigenetic plasticity. Moreover, GSK-J4 has been used to treat not only leukemia but also solid cancers where it exerts anti-proliferative effects (Ntziachristos et al, 2014; Hashizume et al, 2014; Lochmann et al, 2018). Our results strongly provide experimental evidence that targeting JMJD3 elicits anti-cancer effects, at least, in part, through inhibiting cancer stem cell activity.

Methylation and demethylation of H3K27 are important for maintaining stem cell function and determination of proper cell fate. No obvious hematological changes were observed in *Jmjd3^{Δ/Δ}*

cells at steady state (Fig. EV2), and *Jmjd3* deficiency induced limited alterations of global H3K27me3 and gene expression. It is unknown whether other epigenetic factors, such as UTX, another H3K27 demethylase, compensate for loss of JMJD3 to maintain normal hematopoiesis. To address this possibility, we compared RNA-seq data from $Jmjd3^{\Delta/\Delta}$ and $Utx^{\Delta/\Delta}$ LSK cells (Sera et al, 2021) and analyzed genes that were more than 2-fold upregulated or downregulated following *Jmjd3* or *Utx* deficiency. We found no substantial overlap in either upregulated or downregulated genes (Fig. EV6D,E), indicating that the target genes of JMJD3 and UTX are mutually exclusive, suggesting it is unlikely that UTX compensates for JMJD3 deficiency.

In this study, we investigated the roles of JMJD3 in adult hematopoiesis and found that the JMJD3-p16[INK4a] axis mediated cellular senescence functions as a hematopoietic gatekeeper not only to protect HSPCs from excessive cell cycle entry and eventual exhaustion under stress, but also to enhance the stemness in HSPCs exposed to stress via its demethylase activity. Mallaney and colleagues reported that *Jmjd3* deficient mice generated using the Vav-Cre system led to defects on HSC repopulation capacity (Mallaney et al, 2019), similar to our conditional knockout model using the Mx-Cre system. However, their mouse model exhibited reduced HSCs at steady state that was not observed in our study. The reason for the discrepancy is not clear, but one possibility is the timing of loss of JMJD3, namely, fetal hematopoiesis in that study and adult hematopoiesis in this study. Previous reports also demonstrated that the inherited and acquired loss of the target gene displayed different phenotypes (Rathinam et al, 2010; Nakata et al, 2017). Additionally, their model also presented a proliferative advantage of HSCs under inflammatory stress, which was mediated by AP-1 (FOS and JUN) activation. Given that changes in H3K27me3 levels at the *c-Fos* and *c-Jun* loci upon JMJD3 deletion in HSCs were not observed, these findings suggest that the JMJD3-p16[INK4a] axis may regulate HSC repopulating capacity through upstream cellular senescence via H3K27me3 modification, thereby inhibiting AP-1 activation. Our findings provide insights into the regulatory mechanisms of hematopoiesis through histone modifications and suggest that JMJD3 is a prospective target for stem cell aging research and anti-cancer stem cell therapies.

## Methods

### Reagents and tools table

| Reagent/resource | Reference or source | Identifier or catalog number |
|---|---|---|
| *Experimental models* | | |
| *Jmjd3^{flox/flox}* mice | This paper | N/A |
| *MxCre^+* mice | Kühn et al, 1995 | N/A |
| *Jmjd3-Flag* KI mice | This paper | N/A |
| *Recombinant DNA* | | |
| *pMYs-MLL-AF9-IRES-EGFP* | Koide et al, 2016 | N/A |
| *pMYs-mouse p16^{Ink4a}-IRES-KO* (Kusabira Orange) | This paper | N/A |

| Reagent/resource | Reference or source | Identifier or catalog number |
|---|---|---|
| *pMYs-mouse Ccnd1^{WT}-IRES-EGFP* | This Paper | N/A |
| *pMYs-mouse Ccnd1^{T156A}-IRES-EGFP* | This paper | N/A |
| *Oligonucleotides* | | |
| Primers for qPCR | Sequence (5'–3') | |
| *p16^{Ink4a}* | Fw: CCCAACGCCCCGAACT | N/A |
| | Rv: GTGAACGTTGCCCATCATCA | |
| *p15^{Ink4b}* | Fw: TCAGAGACCAGGCTGTAGCAATC | N/A |
| | Rv: CCCCGGTCTGTGGCAGAA | |
| *p18^{Ink4c}* | Fw: AACCATCCCAGTCCTTCTGTCA | N/A |
| | Rv: CCCCTTTCCTTTGCTCCTAATC | |
| *p19^{Ink4d}* | Fw: CGGTATCCACTATGCTTCTGGAA | N/A |
| | Rv: CCGCTGCGCCACTCAA | |
| *p21* | Fw: TTCCGCACAGGAGCAAAGT | N/A |
| | Rv: CGGCGCAACTGCTCACT | |
| *p27* | Fw: GGCCCGGTCAATCATGAA | N/A |
| | Rv: TTGCGCTGACTCGCTTCTTC | |
| *p57* | Fw: CAGCGGACGATGGAAGAACT | N/A |
| | Rv: CTCCGGTTCCTGCTACATGAA | |
| *p19^{Arf}* | Fw: GCTCTGGCTTTCGTGAACAT | N/A |
| | Rv: GTGAACGTTGCCCATCATCA | |
| *Bmi1* | Fw: TGTGTCCTGTGTGGAGGGTA | N/A |
| | Rv: TGGTTTTGTGAACCTGGACA | |
| *Jmjd3* | Fw: CCATCGCTAAATACGCACAGTAC | N/A |
| | Rv: GGCCAATGTTGATGTTGACTGAG | |
| *Hprt* | Fw: GCTGGTGAAAAGGACCTCTCG | N/A |
| | Rv: CCACAGGACTAGAACACCTGC | |
| Primers for ChIP | Sequence (5'–3') | |
| *Cdkn2a #1* | Fw: GTCGCAGGTTCTTGGTCACT | N/A |

| Reagent/resource | Reference or source | Identifier or catalog number |
|---|---|---|
| | Rv: ATGTTCACGAAAGCCAGAGC | |
| *Cdkn2a* #2 | Fw: AAGGGTCAACTGTCCTGTGG | N/A |
| | Rv: TCATACCCAGGGACCTCTTG | |
| *Cdkn2a* #3 | Fw: TCCAGACACACAAATGCACA | N/A |
| | Rv: AACAGGGGAACGGAGAGTTT | |
| *Cdkn2a* #4 | Fw: AATGCCAGGCCTTTAATCCT | N/A |
| | Rv: GAGGCAGGAAGAAGAACACG | |
| *p21* (Neg ctrl) | Fw: GAAGGCTTCGTTTGTTGGAG | N/A |
| | Rv: TCCAAGGACTGGAAGAGTGG | |
| *Antibodies* | | |
| Rat anti-mouse-B220-APC for FACS | BD Pharmingen | Cat#553092 |
| Rat anti-mouse-B220-Biotin for FACS | BD Pharmingen | Cat#559971 |
| Rat anti-mouse-B220- PE-Cy7 for FACS | BD Pharmingen | Cat#552772 |
| Anti-BrdU-APC for FACS | BD Pharmingen | Cat#552598 |
| Anti-Annexin V-APC for FACS | BD Pharmingen | Cat#550475 |
| Rat anti-mouse-c-kit-APC-H7 for FACS | BD Pharmingen | Cat#560185 |
| Anti-mouse-c-kit-beads for MACS | Miltenyi Biotec | Cat#130-097-146 |
| Rat anti-mouse-CD135-PE for FACS | BD Pharmingen | Cat#553842 |
| Anti-mouse-CD16/32-BV510 for FACS | BD Pharmingen | Cat#740111 |
| Rat anti-mouse-CD16/32-PE for FACS | BD Pharmingen | Cat#553145 |
| Rat anti-mouse-CD34-BV421 for FACS | BD Pharmingen | Cat#562608 |
| Rat anti-mouse-CD34-eFluor660 for FACS | eBioscience | Cat#50-0341-82 |
| Rat anti-mouse-CD34-FITC for FACS | BD Pharmingen | Cat#553733 |
| Rat anti-mouse-CD4-Biotin for FACS | eBioscience | Cat#13-0042-82 |
| Rat anti-mouse-CD8a-Biotin for FACS | BD Pharmingen | Cat# 553029 |
| Anti-mouse-Gr-1-APC for FACS | Biolegend | Cat#108412 |
| Rat anti-mouse-Gr-1-Biotin for FACS | BD Pharmingen | Cat#559971 |
| Rat anti-mouse-Gr-1-PE-Cy7 for FACS | BD Pharmingen | Cat#565033 |
| Mouse anti-mouse-Ly5.1-APC for FACS | BD Pharmingen | Cat#558701 |
| Mouse anti-mouse-Ly5.2-PE for FACS | BD Pharmingen | Cat#560695 |
| Rat anti-mouse-Mac-1-Biotin for FACS | BD Pharmingen | Cat#559971 |
| Rat anti-mouse-Mac-1-FITC for FACS | BD Pharmingen | Cat#553310 |
| Rat anti-mouse-Sca-1-PE-Cy7 for FACS | BD Pharmingen | Cat#558162 |
| Anti-Streptavidin-PerCP-Cy5.5 for FACS | BD Pharmingen | Cat#551419 |
| Anti-Streptavidin-beads for MACS | Miltenyi Biotec | Cat#130-048-101 |
| Rat anti-mouse-Ter119-Biotin for FACS | BD Pharmingen | Cat#559971 |
| Rat Anti-mouse-Thy1.2-FITC for FACS | BD Pharmingen | Cat#553004 |
| Mouse Anti-β-ACTIN for WB | Sigma Aldrich | Cat#A5316 |
| Mouse anti-α-TUBULIN for WB | Sigma Aldrich | Cat#T5168 |
| Mouse anti-BMI1 for ChIP | Active Motif | Cat#39993 |
| Rabbit anti-H3K27me3 for IF, ChIP and CUT&RUN | Cell Signaling | Cat#9733 |
| Rabbit anti-FLAG for WB and ChIP | Sigma | Cat#F3165 |
| Rabbit anti-JMJD3 for WB | Cell Signaling | Cat#3457 |
| Rabbit anti-p16$^{INK4a}$ for WB | Cell Signaling | Cat#4824 |
| Rabbit anti-IgG for CUT&RUN | Cell Signaling | Cat#66362 |
| *Bacterial and virus strains* | | |
| Stbl2 Competent cells | Thermo Fisher | Cat#10268019 |
| *Chemicals, peptides, and recombinant proteins* | | |
| GSK-J4 | MCE MedChemExpress | Cat#HY-15648B |
| Polyinosinic-polycytidylic acid (pIpC) | Sigma | Cat#P1530 |
| Hoechst 33342 | Dojindo Laboratories | Cat#346-07951 |

| Reagent/resource | Reference or source | Identifier or catalog number |
|---|---|---|
| BrdU | BD Biosciences | Cat#552598 |
| Adriamycin | Wako | Cat#040-21521 |
| Z-VAD-FMK | Selleck | Cat#S7023 |
| Recombinant murine SCF | Peprotech | Cat#250-03 |
| Recombinant murine IL-3 | Peprotech | Cat#213-13 |
| Recombinant murine IL-6 | Peprotech | Cat#216-16 |
| Recombinant murine GM-CSF | Peprotech | Cat#250-05 |
| Recombinant murine TPO | Peprotech | Cat#315-14 |
| Recombinant murine FLT3-L | Peprotech | Cat#250-31 L |
| p16-TAT peptide | Prospec | Cat#PKA-337 |
| *Critical commercial assays* | | |
| ChIP-IT Express Enzymatic Kit | Active Motif | Cat#53009 |
| CUT&RUN assay kit | Cell Signaling | Cat#86652 |
| NEBNext Ultra II DNA Library Prep Kit | NEB | Cat#E7370L |
| Senescence β-Galactosidase Staining Kit | Cell Signaling | Cat#9860 |
| TruSeq RNA Sample Preparation Kit v2 | Illumina | Cat#RS-122-2001 |
| SureSelect Strand-Specific mRNA Library Preparation kit | Agilent Technologies | Cat#G9691B |
| BrdU Flow Kit | BD Biosciences | Cat#552598 |
| SuperScript VILO master mix kit | Invitrogen | Cat#11755050 |
| *Software* | | |
| FlowJo software | BD Biosciences | https://www.flowjo.com/solutions/flowjo |
| ImageJ64 | National Institutes of Health | https://imagej.nih.gov/ij/index.html |
| Enrichr | Department of Pharmacological Sciences, Icahn School of Medicine at Mount Sinai | https://maayanlab.cloud/Enrichr/ |
| GSEA | Broad Institute | https://www.gsea-msigdb.org/gsea/index.jsp |
| Genome Browser | University of California Santa Cruz | https://genome.ucsc.edu |
| Transcription Factor and Histone ChIP-Seq pipeline | ENCODE | https://github.com/ENCODE-DCC/chip-seq-pipeline2 |

| Reagent/resource | Reference or source | Identifier or catalog number |
|---|---|---|
| Homer v4.11 | University of California San Diego | http://homer.ucsd.edu/homer/ |
| GraphPad Prism version 10 | GraphPad | https://www.graphpad.com |

## Generation of *Jmjd3* conditional knockout mice

A bacterial artificial chromosome clone containing the mouse *Jmjd3* gene was purchased from the BACPAC Resource Center, Children's Hospital Oakland Research Institute (Oakland). Fragments from the Aor51HI site in intron 11 to the artificially introduced BamHI site in intron 14 and from the SnaBI site in intron 17 to the BamHI site in intron 20 were used as the 5′ and 3′ arms of the targeting vector, respectively. A fragment from the artificially introduced BamHI site in intron 14 to the SnaBI site in intron 17 that contains exons 15–17 was *floxed* and inserted between the two arms, together with an *Frt*-flanked *Neomycin* (*Neo*)-resistance gene. A *diphtheria toxin A* (*DTA*) gene was attached to the 5′ end of the vector as a negative selector. KY1.1 ES cells (kindly provided by Dr. Junji Takeda, Osaka University, Japan) were electroporated with the linearized vector. Individual clones were screened via 5′ Southern blot using a 5′ probe and 3′ genomic PCR using P1 and P2 primers. Correctly targeted ES cells were microinjected into the blastocysts derived from C57BL/6 × BDF1 mice, and the resultant chimeric male mice were crossed with C57BL/6 female mice to transmit the mutant allele to progeny. The *Neo* resistance gene was removed by crossing heterozygotes with CAG-FLPe transgenic mice (RIKEN BRC, RBRC1834). The resultant *Jmjd3*$^{+/flox}$ mice were crossed to *Mx-1Cre* mice, and the activation of Cre was induced by an intraperitoneal injection of polyinosinic-polycytidylic acid (pIpC). Mice that were backcrossed to C57BL/6 N for at least five generations were used for experiments. Mouse experiments were performed in strict accordance with the Guide for the Care and Use of Laboratory Animals of the Hiroshima University Animal Research Committee (permission No. 29-111) and the Institute for Laboratory Animals of Tokyo Women's Medical University Animal Research Committee (permission No. AE23-065).

## Generation of *Jmjd3-Flag* knock-in mice

Generation of *Jmjd3-Flag* KI mice was performed as previously described (Aida et al, 2015 and Quadros et al, 2017). To insert 3× Flag sequences (5′-GACTATAAGGATCACGATGGAGATTATAAGGACCATGATATAGACTACAAAGACGATGACGACAAA-3′ encoding DYKDHDGDYKDHDIDYKDDDDK) upstream of the stop codon of *Jmjd3*, a CRISPR RNA (crRNA) was designed to cut one base upstream of the stop codon, and a single strand oligo donor nucleotide (ssODN) composed of 5′ arm (250 bases) (Fasmac), 3× Flag sequences and 3′ arm (250 bases) (Fasmac) was microinjected into pronuclei of fertilized eggs of C57BL6/N mice, together with the crRNA (Fasmac), a trans-activating crRNA (Fasmac) and Cas9 protein (New England Biolabs Japan). The

*KI* illustration is shown in Fig. 3I. All animal care and experimental procedures were conducted in accordance with the guidelines of the animal ethics committees of Tokyo Women's Medical University (AE24-101).

## Flow cytometric analysis

Mononuclear cells (MNCs) were isolated from BM and PB cells. After incubation with Fc-blocker (BD Biosciences), MNCs were stained with antibodies. The surface marker phenotypes of each hematopoietic fraction are summarized in Table EV1. Flow cytometry was performed on a FACSCanto II and a FACSAria II, and data were analyzed with the FlowJo software (BD Biosciences).

## BMT assay

*Jmjd3*$^{+/+}$ or *Jmjd3*$^{\Delta/\Delta}$ HSPCs (Ly5.2), together with competitor BM MN cells (Ly5.1) for radioprotection, were transplanted intravenously into recipient mice (Ly5.1) lethally irradiated at a dose of 8.0 Gy, as previously described (Nakata et al, 2017). The chimerism of donor-derived hematopoietic cells was monitored by flow cytometry.

## Western blot

Whole-cell lysates were run on SDS-PAGE gels and transferred onto a PVDF membrane (Millipore). Non-specific binding was blocked with 5% skim milk in Tris-buffered saline with 0.1% Tween-20 (TBS-T), followed by an overnight reaction with primary antibodies. Immunoreactive proteins were identified with HRP-conjugated anti-rabbit IgG (GE Healthcare) and anti-mouse IgG (GE Healthcare), and reacted with chemiluminescent HRP substrate (Millipore).

## Retroviral transduction

LK cells were first stained by anti-lineage-biotin and enriched by MACS LS-columns, streptavidin microbeads, and anti-c-kit microbeads. For transduction of *MLL-AF9* and/or *p16*$^{Ink4a}$, LK cells were cultured for 3 days in RPMI1640 medium containing 10% FCS supplemented with 20 ng/ml SCF, 10 ng/ml IL-3, and 10 ng/ml IL-6. The cells were infected with retrovirus carrying *MLL-AF9*-IRES-*EGFP* and/or *p16*$^{Ink4a}$-IRES-*KO* (*Kusabira Orange*). For transduction of *Ccnd1*, LK cells were cultured for 3 days in S-clone SF-O3 serum-free medium (EIDIA Co) containing 1% BSA supplemented with 100 ng/ml SCF and 100 ng/ml TPO, and the cells were transduced with retrovirus carrying mouse *Ccnd1*$^{WT}$- or mouse *Ccnd1*$^{T156A}$-IRES-*EGFP*.

## Colony formation and serial replating assays

Cells were seeded in MethoCult™ M3231 (STEMCELL Technologies), supplemented with cytokines (PeproTech). For normal HSPCs, 50 ng/ml SCF, 50 ng/ml TPO, and 50 ng/ml FLT3L were used. For *MLL-AF9*-transformed LK cells, 20 ng/ml SCF, 10 ng/ml IL-3, 10 ng/ml IL-6, and 10 ng/ml GM-CSF were used. Colony numbers were counted at 4–6 days after plating, and the cells were collected and replated.

## Immunofluorescent staining

Cells were plated on poly-L-lysine-coated glass slides (Matsunami Glass), fixed with 4% paraformaldehyde and permeabilized with 0.5% Triton X-100. Non-specific binding was blocked with Protein block serum-free (Dako) and cells were stained with the primary antibody. After washing with phosphate-buffered saline (PBS), the cells were stained with Alexa Fluor 488- or 555-conjugated goat anti-rabbit IgG antibody (Invitrogen). Nuclei were counterstained with Hoechst 33342. Images were taken with an FV-1000 confocal microscope (Olympus). Cells were chosen randomly in multiple fields, and fluorescence intensities of individual cells (more than 110 cells per experiment) were quantified computationally using ImageJ64 software.

## Quantitative real-time PCR

Total cellular RNA was isolated with TRIzol reagent (Invitrogen). RNA was reverse-transcribed with a SuperScript VILO master mix kit (Invitrogen), according to the manufacturer's protocol. Quantitative real-time PCR was performed with power SYBR Green PCR Master Mix (Thermo Fisher Scientific) on a StepOnePlus Real-Time PCR system (Applied Biosystems) running StepOne 2.3 Software (Applied Biosystems). All data are presented relative to *Hprt*.

## Transcriptome analysis and data processing

RNA-Seq libraries were prepared with the TruSeq RNA Sample Preparation Kit v2 and SureSelect Strand-Specific mRNA Library Preparation kit. Transcriptome analysis was performed using a next-generation sequencer (GAIIx and HiSeq 2500; Illumina), according to the manufacturer's instructions. The sequence tags (more than $3 \times 10^7$ reads for each sample) were mapped onto the mouse genomic sequence (UCSC Genome Browser, version mm10) with ELAND for DRA004290 and TopHat for DRA008581, DRA005628 and DRA010428. Normalized gene expression was compared between *Jmjd3*$^{\Delta/\Delta}$ and *Jmjd3*$^{+/+}$ LSK cells at steady state, LSK cells after BMT, and L-GMPs, and between DMSO and GSK-J4-treated L-GMPs, and the results were analyzed with GSEA software. Gene sets with a false discovery rate (FDR) *q*-value < 0.25 were considered statistically significant.

## Analysis of cell cycle activity

BrdU incorporation was analyzed with a BrdU Flow Kit. *Jmjd3*$^{+/+}$ or *Jmjd3*$^{\Delta/\Delta}$ mice were intraperitoneally injected with 1 mg of BrdU at 8, 16, and 24 h before analysis. Cultured LK cells transfected with *MLL-AF9* were treated with BrdU (10 µM) in cultured medium 8 h before analysis. Cells were stained with antibodies of cell surface markers and then fixed, permeabilized, and stained with an anti-BrdU antibody according to the manufacturer's instructions.

## Analysis of apoptosis

*Jmjd3*$^{+/+}$ or *Jmjd3*$^{\Delta/\Delta}$ cells were stained with antibodies of cell surface markers then resuspended in binding buffer (10 mM HEPES/NaOH (pH 7.4) 140 mM NaCl, 2.5 mM CaCl$_2$). Cells were

re-stained with an anti-Annexin V antibody according to the manufacturer's instructions.

## ChIP-qPCR assay

Chromatin was enzymatically shred with the ChIP-IT Express Enzymatic Kit and immunoprecipitated overnight at 4 °C with Protein G magnetic beads and antibodies. Subsequently, chromatin was eluted, reverse cross-linked, and treated with Proteinase K, according to the manufacturer's instructions. ChIP-qPCR data are presented as relative enrichment levels normalized to the *p21* promoter region as a negative control (Neg ctrl).

## CUT&RUN assay and analysis

Anti-H3K27me3 and anti-IgG were used for CUT&RUN. This assay was performed with the CUT&RUN assay kit, according to the manufacturer's protocol. Libraries were generated using the NEBNext Ultra II DNA Library Prep Kit for Illumina and sequenced on an Illumina NovaSeq 6000. Paired-end fastq files were processed using the ENCODE Transcription Factor and Histone ChIP-Seq pipeline. Reads were trimmed using cutadapt v2.5. and aligned to the mm10 genome using Bowtie2 v2.3.4.3., and SAMtools v1.9 was used to convert the output file to BAM format. Duplicates were removed using Picard Tools v2.20.7. Peak calling was performed with MACS2 v2.2.4, and the peaks were compared with IgG peaks before subsequent analysis. Bigwig files were normalized by a scaling factor based on the read counts of spike-in DNA using Deeptools v3.3.1 bamCoverage tools and visualized in the UCSC genome browser. Homer v4.11 was used for peak annotation analysis. Bedtools v2.29.0 intersect was used to determine peak overlaps and assign target genes.

## β-galactosidase staining assay

Senescence-associated β-galactosidase staining was performed using a Senescence β-Galactosidase Staining Kit. *Jmjd3*[+/+] and *Jmjd3*[Δ/Δ] LSK cells at steady state, LSK cells after BMT, and L-GMPs were cultured with 0.025 μg/ml of adriamycin (ADR) and 10 μM of Z-VAD-FMK (caspase inhibitor). The cells were washed with PBS, then fixed and stained with β-galactosidase substrate X-Gal according to the manufacturer's instructions. The senescent cells were counted using ImageJ software.

## p16[INK4a]-TAT treatment

LK cells were cultured for 4 days in S-clone SF-O3 serum-free medium containing 1% BSA supplemented with cytokines (100 ng/ml SCF, 100 ng/ml TPO, 100 ng/ml FLT3L, and 50 ng/ml IL-6). BSA was then added to the cultured medium or 50 nM of p16-TAT. Fresh BSA or p16-TAT (to 10% of total culture medium) was added every 24 h. After incubation for 4 days, cells were subject to colony formation and BMT assays.

## GSK-J4 treatment

*MA9* cells were exposed to 5 or 10 μM of GSK-J4 for 24 h. After incubation for 24 h, cells were subject to colony formation and BMT assays. For GSK-J4 treatment in vivo, *MA9*-expressing cells (Ly5.2) were transplanted intravenously into

recipient mice (Ly5.1) lethally irradiated at a dose of 8.0 Gy with $2.5 \times 10^5$ wild-type competitor MNBM cells for radioprotection. Ten days after BMT, DMSO or GSK-J4 (50 mg/kg/day) was intraperitoneally injected into the recipients for 5 consecutive days.

## 5-FU treatment

5-FU (150 mg/kg) was intraperitoneally injected into *Jmjd3*[+/+] and *Jmjd3*[Δ/Δ] mice as described (Takubo K et al, 2023).

## Statistical analysis

In all experiments, virtually identical sample sizes per group were adjusted to avoid experimental bias. Statistical differences between the means of two groups were assessed using two-tailed Student's *t* test. When multiple treatment groups were compared, statistical significance of differences was assessed using a one-way analysis of variance followed by Dunnett's test. Mouse survival curves were constructed using the Kaplan–Meier methodology and compared with a log-rank test. All statistical tests were performed with GraphPad Prism version 10. In quantitative PCR, SDs were calculated with technical triplicates, thus statistical significance was not shown.

# Data availability

All sequencing data has been deposited in the DNA Data Bank of Japan. RNA-seq; DRA004290 (https://ddbj.nig.ac.jp/resource/sra-submission/DRA004290), DRA008581 (https://ddbj.nig.ac.jp/resource/sra-submission/DRA008581), DRA005628 (https://ddbj.nig.ac.jp/resource/sra-submission/DRA005628), and DRA010428 (https://ddbj.nig.ac.jp/resource/sra-submission/DRA010428). CUT&RUN-seq; DRA020409 (https://ddbj.nig.ac.jp/resource/sra-submission/DRA020409) and DRA016982 (https://ddbj.nig.ac.jp/resource/sra-submission/DRA016982). The source data of this paper are collected in the following database record: biostudies:S-SCDT-10_1038-S44319-025-00502-9.

# Peer review information

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

## Acknowledgements

We thank Yuki Sakai, Sawako Ogata, Megumi Nakamura, Mika Takeda, Ayako Kawabata, and Masafumi Yamanaka for animal care, genotyping, and molecular experiments. We also thank Institute for Institute of Laboratory Animals (ILA) and Comprehensive Medical Sciences (ICMS) of Tokyo Women's Medical University for experimental support. This work was in part supported by Grant-in-Aid for Japan Society for the Promotion of Science Fellows, The Takeda Science Foundation, The Naito Foundation, and AMED-CREST (25gm131006).

## Author contributions

**Yuichiro Nakata**: Conceptualization; Data curation; Formal analysis; Supervision; Funding acquisition; Investigation; Writing—review and editing. **Takeshi Ueda**: Conceptualization; Data curation; Supervision. **Yasuyuki Sera**: Formal analysis. **Miho Koizumi**: Data curation; Formal analysis. **Katsutoshi Imamura**: Formal analysis. **Akinori Kanai**: Data curation; Formal analysis. **Ken-ichiro Ikeda**: Formal analysis. **Norimasa Yamasaki**: Conceptualization; Resources. **Akiko Nagamachi**: Resources; Formal analysis. **Kohei Kobatake**: Formal analysis; Investigation. **Masataka Taguchi**: Conceptualization; Investigation. **Yusuke Sotomaru**: Resources; Investigation. **Tatsuo Ichinohe**: Investigation. **Zen-ichiro Honda**: Conceptualization; Resources; Supervision; Investigation. **Takuro Nakamura**: Formal analysis; Supervision. **Ichiro Manabe**: Formal analysis; Investigation. **Toshio Suda**: Supervision; Investigation. **Keiyo Takubo**: Conceptualization; Resources; Methodology. **Osamu Kaminuma**: Conceptualization; Resources; Supervision; Methodology. **Hiroaki Honda**: Conceptualization; Resources; Supervision; Funding acquisition; Project administration; Writing—review and editing.

Source data underlying figure panels in this paper may have individual authorship assigned. Where available, figure panel/source data authorship is listed in the following database record: biostudies:S-SCDT-10_1038-S44319-025-00502-9.

## Disclosure and competing interests statement

The authors declare no competing interests.

# Expanded View Figures

A

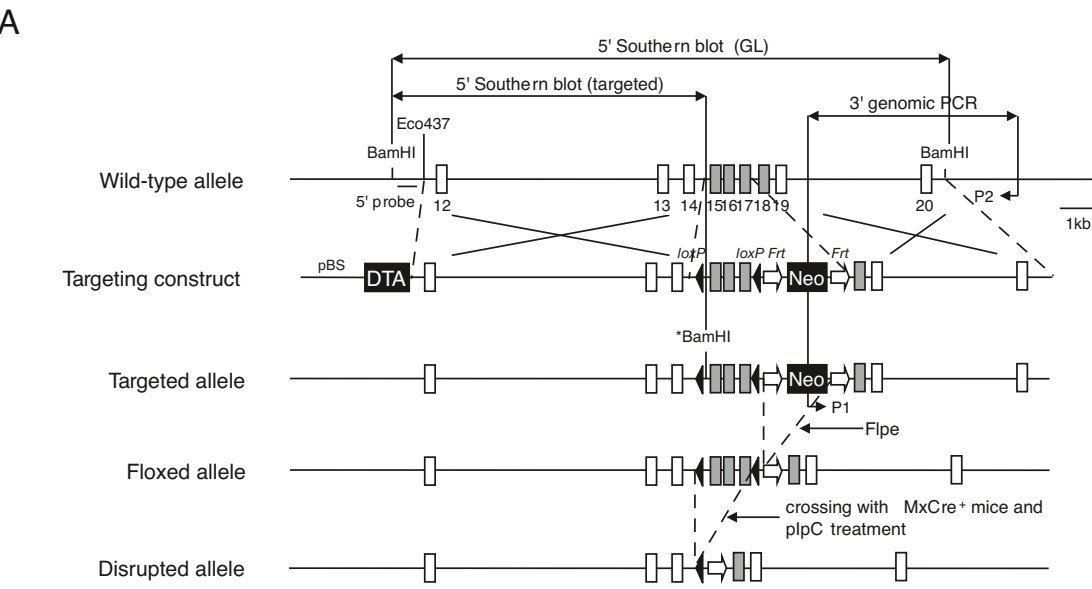

B

C

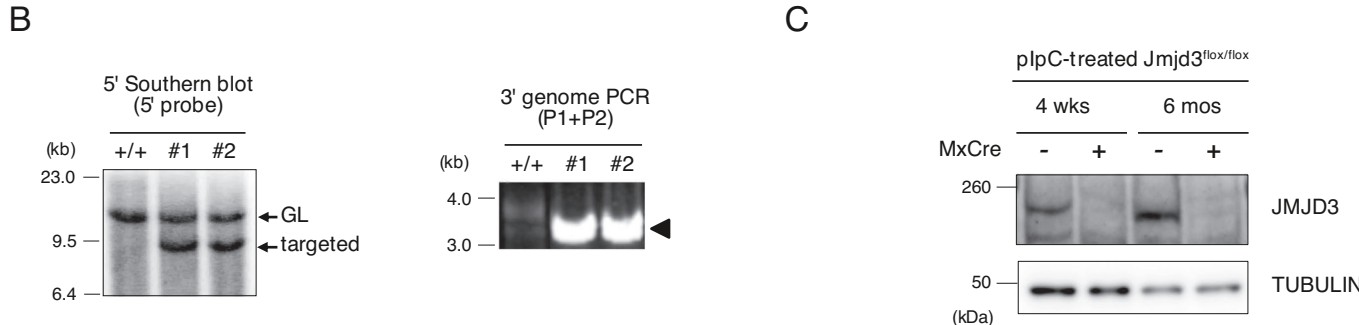

**Figure EV1. Targeting strategy, genotyping of ES clones, and deletion of *Jmjd3*.**

(A) Exons 15–17 of the mouse *Jmjd3* gene were encompassed by two *loxP* sites (black triangles), and a *neomycin*-resistance gene (*Neo*) was flanked by two *Frt* sites (white arrows). After removing *Neo* by Flpe, *floxed* exons were deleted by crossing with *MxCre*⁺ mice and pIpC treatment. The position of the genomic probe for the 5′ Southern blot (5′ probe), primers for the 3′ genomic PCR (P1 and P2), and the positions of restriction enzymes (BamHI and Eco437) are shown. BamHI with an asterisk (*BamHI) is an artificial enzyme site introduced by in vitro mutagenesis. Gray boxes indicate the exons that encode the JmjC domain. (B) Homologously recombined ES clones (#1 and #2) identified by 5′ Southern blot and 3′ genomic PCR. Germline (GL) and targeted bands in 5′ Southern blot are indicated by arrows (left panel) and PCR products for the 3′ genomic PCR are indicated by arrowheads (right panel). (C) Immunoblot showing JMJD3 protein in bone marrow (BM) cells of *Jmjd3^flox/flox*; *MxCre*⁺ and *Jmjd3^flox/flox*; *MxCre*⁻ mice at 4 weeks (4 wks) and 6 months (6 mos) after pIpC treatment.

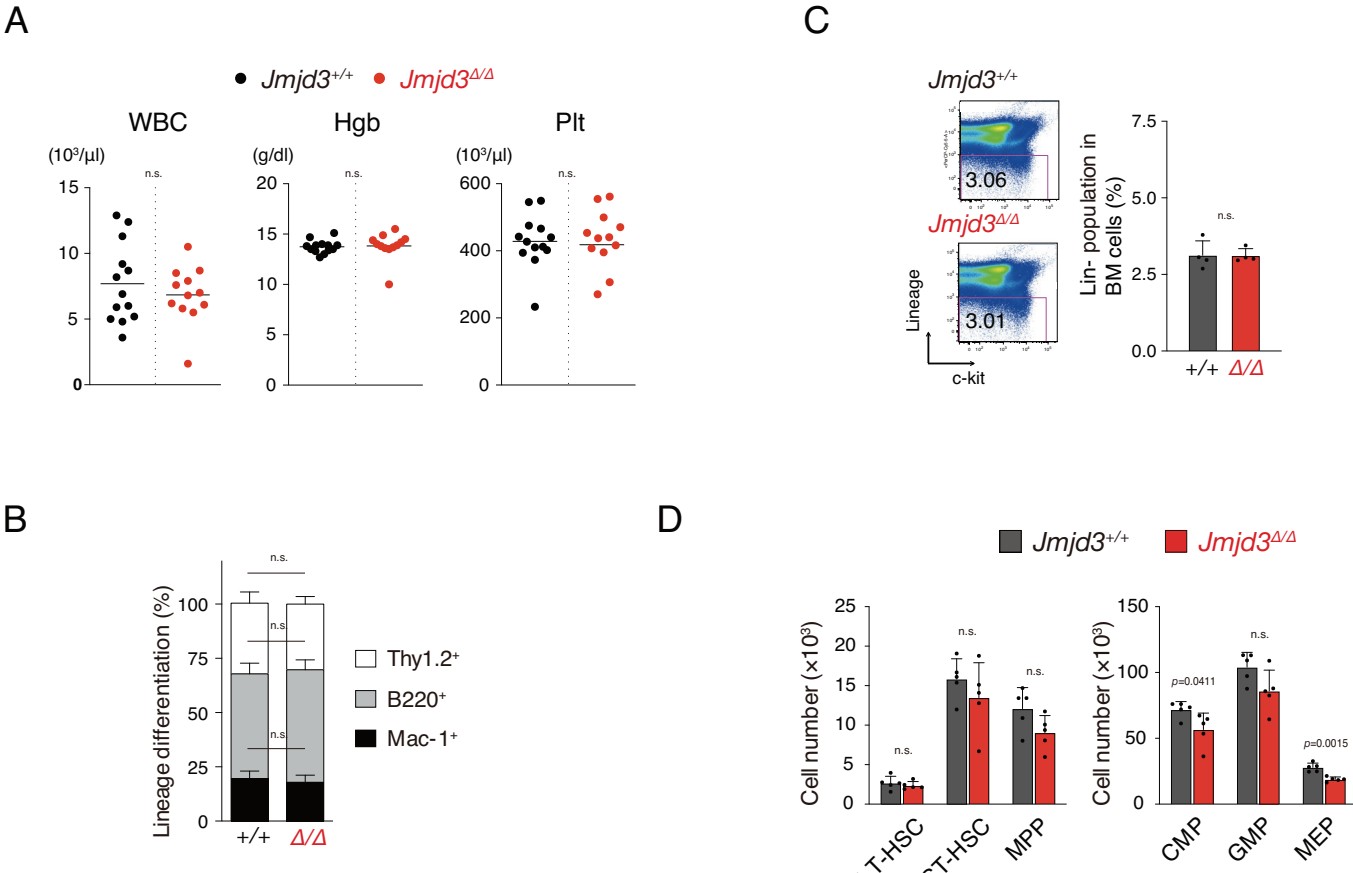

**Figure EV2. Analysis of *Jmjd3^{Δ/Δ}* hematopoietic cells at steady state.**

(A) Analysis of peripheral blood (PB) parameters in *Jmjd3^{+/+}* and *Jmjd3^{Δ/Δ}* mice at 4 weeks after pIpC treatment. White blood cell (WBC) counts, hemoglobin concentration (Hgb), and platelet (Plt) number in the PB of *Jmjd3^{+/+}* ($n = 13$) and *Jmjd3^{Δ/Δ}* mice ($n = 12$) are plotted as dots, and the mean values are indicated as bars. Student's *t* test was used to calculate *p* value. (B) Analysis of lineage differentiation (Thy1.2^+, B220^+, and Mac-1^+ cells) in the PB cells of *Jmjd3^{+/+}* ($n = 13$) and *Jmjd3^{Δ/Δ}* mice ($n = 12$) (mean ± SD). Student's *t* test was used to calculate *p* value. (C) Flow cytometric profiles of lineage^- (Lin^-) cells in the BM of *Jmjd3^{+/+}* and *Jmjd3^{Δ/Δ}* mice (mean + SD, $n = 5$). Student's *t* test was used to calculate *p* value. (D) Absolute numbers of HSPC subpopulations (LT-HSC, ST-HSC, and MPP) and myeloid progenitors (CMP, GMP, and MEP) in the BM of *Jmjd3^{+/+}* and *Jmjd3^{Δ/Δ}* mice (mean + SD, $n = 5$). Student's *t* test was used to calculate *p* values.

| Gene set | LSK (Steady) ($\Delta/\Delta$ vs +/+) | | LSK (BMT) ($\Delta/\Delta$ vs +/+) | | L-GMP ($\Delta/\Delta$ vs +/+) | |
|---|---|---|---|---|---|---|
| | NES | FDR q | NES | FDR q | NES | FDR q |
| PRC2_EZH2_UP.V1_UP | -0.88 | 0.973 | -0.83 | 0.809 | -1.16 | 0.218 |
| PRC2_EED_UP.V1_UP | -1.04 | 0.345 | -1.31 | 0.059 | -1.61 | 0.003 |
| PRC2_SUZ12_UP.V1_UP | -1.07 | 0.297 | 1.01 | 0.446 | -1.23 | 0.147 |
| PRC1_BMI_UP.V1_UP | -0.86 | 0.855 | -1.29 | 0.090 | -1.55 | 0.013 |

(Bracken, AP., *et al*, *Genes Dev*, 2006)

| Gene set | | | | | | |
|---|---|---|---|---|---|---|
| BMI_UP.V1_UP | 1.02 | 0.379 | -1.08 | 0.349 | -1.69 | 0.004 |

(Wiederschain, D., *et al*, *Mol Cell Biol*, 2007)

| Gene set | | | | | | |
|---|---|---|---|---|---|---|
| DOUGLAS_BMI1_TARGETS_UP | 1.02 | 0.402 | -1.14 | 0.123 | -0.91 | 0.702 |

(Douglas, D., *et al*, *Cancer Res*, 2008)

| Gene set | | | | | | |
|---|---|---|---|---|---|---|
| BENPORATH_PRC2_TARGETS | -0.36 | 1 | -1.49 | 0.014 | -1.78 | 0.003 |

(Ben-Porath, I., *et al*, *Nat Genet*, 2008)

| Gene set | | | | | | |
|---|---|---|---|---|---|---|
| KONDO_EZH2_TARGETS | 0.90 | 0.723 | -1.02 | 0.446 | -1.45 | 0.027 |

(Kondo Y., *et al*, *Nat Genet*, 2008)

**Figure EV3.   JMJD3 competitively regulates Polycomb targets under stress.**

Comparison of gene set enrichment between Polycomb proteins and JMJD3. Gene sets upregulated in cells deficient in the indicated Polycomb genes were compared with those downregulated in *Jmjd3*-deficient LSK (Steady), LSK (BMT), or L-GMP. NES and FDR are indicated. Blue boxes show significantly enriched pathways (FDR < 0.25).

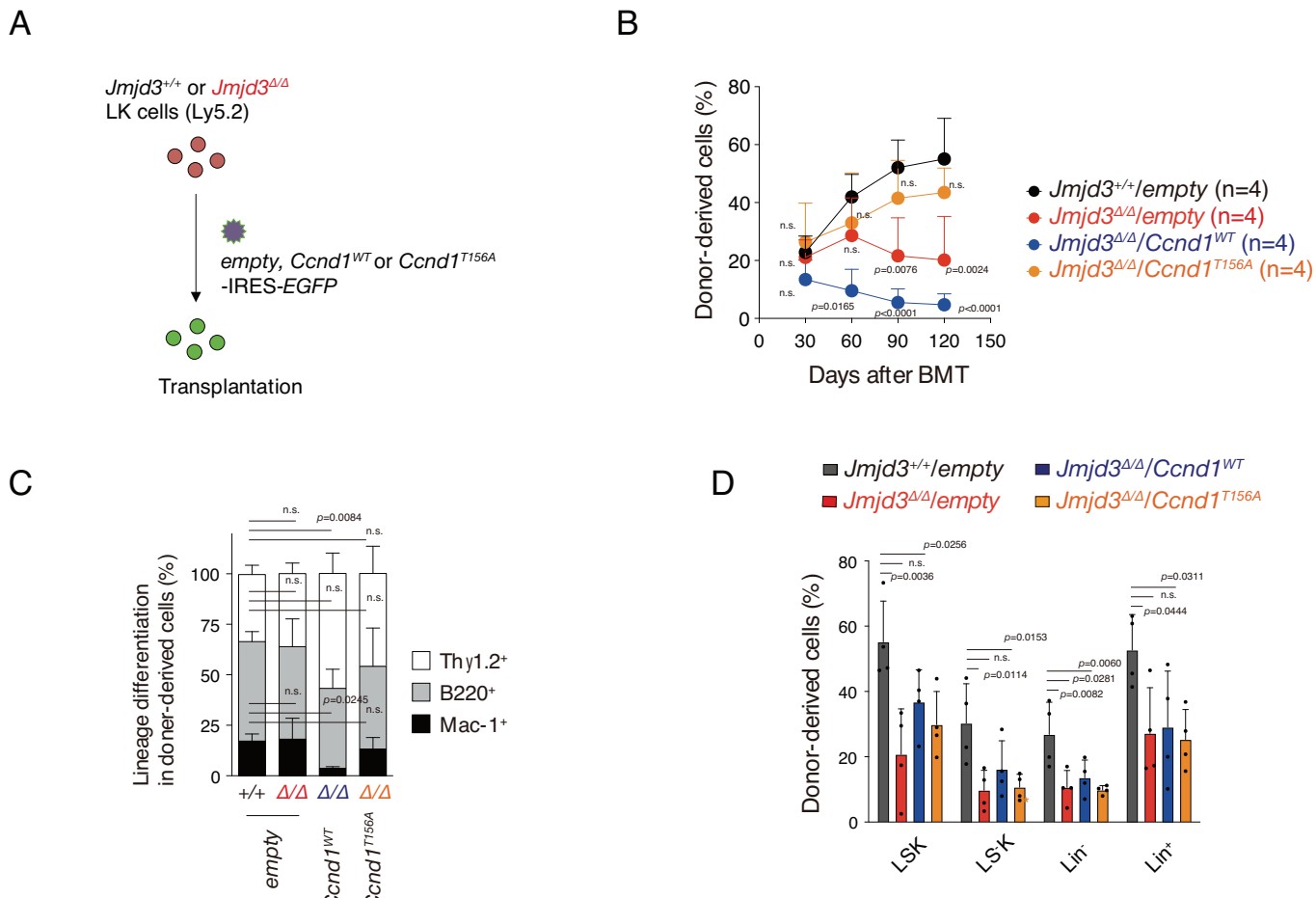

**Figure EV4. HSPC defects caused by *Jmjd3* loss under replicative stress are rescued by retroviral expression of a *Ccnd1* mutant.**

(A) A schematic diagram of *empty*, *Ccnd1*$^{WT}$, and *Ccnd1*$^{T156A}$ transduction. LK cells from *Jmjd3*$^{+/+}$ and *Jmjd3*$^{Δ/Δ}$ mice were transfected with *empty*-, *Ccnd1*$^{WT}$-, or *Ccnd1*$^{T156A}$-IRES-*EGFP* retrovirus, and $5.0 \times 10^4$ EGFP$^+$ cells were transplanted into lethally irradiated recipients with $2.5 \times 10^5$ wild-type competitor MNBM cells for radioprotection (mean + SD, n = 4). (B) Chimerism of donor-derived PB cells in the recipients. Results of mice transplanted with *Jmjd3*$^{+/+}$/*empty*, *Jmjd3*$^{Δ/Δ}$/*empty*, *Jmjd3*$^{Δ/Δ}$/*Ccnd1*$^{WT}$, and *Jmjd3*$^{Δ/Δ}$/*Ccnd1*$^{T156A}$ cells are shown (mean ± SD, n = 4). Dunnett's test was used to calculate *p* value. (C) Lineage differentiation in the donor derived PB cells in recipients 4 months after BMT (mean + SD, n = 4). Dunnett's test was used to calculate *p* value. (D) Chimerism of donor-derived BM cells in recipients 4 months after BMT (mean + SD, n = 4). Dunnett's test was used to calculate *p* values.

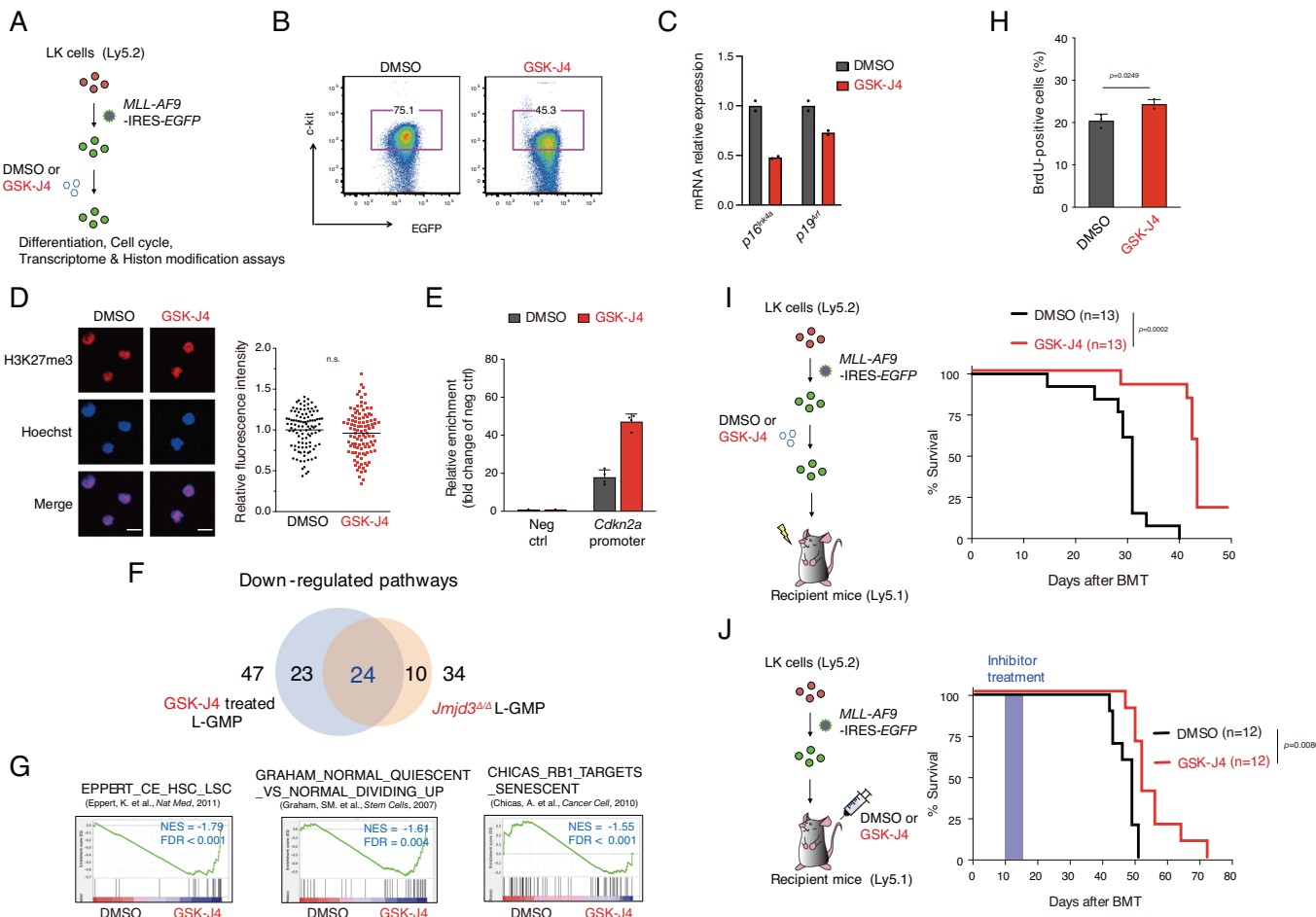

**Figure EV5. Inhibition of JMJD3 suppresses LSC potential by regulating *p16^Ink4a* expression.**

(**A**) Schematic diagram of GSK-J4 treatment after *MLL-AF9* (*MA9*) transduction. LK cells from wild-type mice were transfected with the *MLL-AF9*-IRES-*EGFP* retrovirus. EGFP$^+$ cells were further exposed to DMSO or GSK-J4 for 24 h and subjected to the following assays. (**B**) Flow cytometric profiles of c-kit$^+$ fractions in *MA9* cells treated with DMSO or GSK-J4 (5 or 10 μM) for 24 h. (**C**) qPCR analysis of *Cdkn2a* genes in L-GMPs exposed to DMSO or GSK-J4 (10 μM) for 24 h. (mean + SD, $n = 3$). (**D**) Immunofluorescence staining (left panel) and relative fluorescence intensity (right panel) of H3K27me3 in L-GMPs exposed to DMSO or GSK-J4 (10 μM) for 24 h. Mean values are indicated as bars ($n = 105$). Student's $t$ test was used to calculate $p$ value. Scale bar, 10 μm. (**E**) H3K27me3 levels in the promoter region of *Cdkn2a* (see Fig. 3D) in L-GMPs exposed to DMSO or GSK-J4 (10 μM) for 24 h. Results are shown as fold changes relative to a negative control (Neg ctrl) (mean + SD, $n = 3$). (**F**) Venn diagrams showing the overlap of negatively enriched KEGG pathways in L-GMPs exposed to GSK-J4 (10 μM) for 24 h and *Jmjd3*$^{Δ/Δ}$ L-GMPs. The overlapped pathways are listed in Table EV2. (**G**) GSEA plots of L-GMPs exposed to DMSO or GSK-J4 (10 μM) for 24 h in the indicated gene sets (left, genes commonly upregulated in human HSC and LSC; middle, genes commonly upregulated in quiescent human CD34$^+$ hematopoietic cells; right, genes commonly upregulated through the p16$^{INK4a}$/RB1 pathway). Results are shown with NES and FDR values. (**H**) Flow cytometric analysis of BrdU incorporation in L-GMPs exposed to DMSO or GSK-J4 (5 or 10 μM) for 24 h. (mean + SD, $n = 3$). Student's $t$ test was used to calculate $p$ value. (**I**) Schematic diagram of in vitro GSK-J4 (10 μM) treatment for 24 h after *MLL-AF9* transduction (left panel) and Kaplan–Meier survival plots of mice transplanted with these cells (right panel). In all, $1.0 \times 10^5$ *MA9* cells (Ly5.2$^+$) were transplanted into lethally irradiated recipients with $2.5 \times 10^5$ wild-type competitor MNBM cells ($n = 13$). A log-rank test was used to calculate $p$ value. (**J**) Schematic diagram of in vivo GSK-J4 treatment after *MLL-AF9* transduction (left panel) and Kaplan–Meier survival plots of mice transplanted with *MA9* cells and treated in vivo (right panel). In all, $1.0 \times 10^5$ *MA9* cells (Ly5.2$^+$) were transplanted into lethally irradiated recipients with $2.5 \times 10^5$ wild-type competitor MNBM cells. 10 days after BMT, DMSO or GSK-J4 (50 mg/kg/day) was intraperitoneally injected into the recipients for 5 consecutive days ($n = 12$). A log-rank test was used to calculate $p$ values.

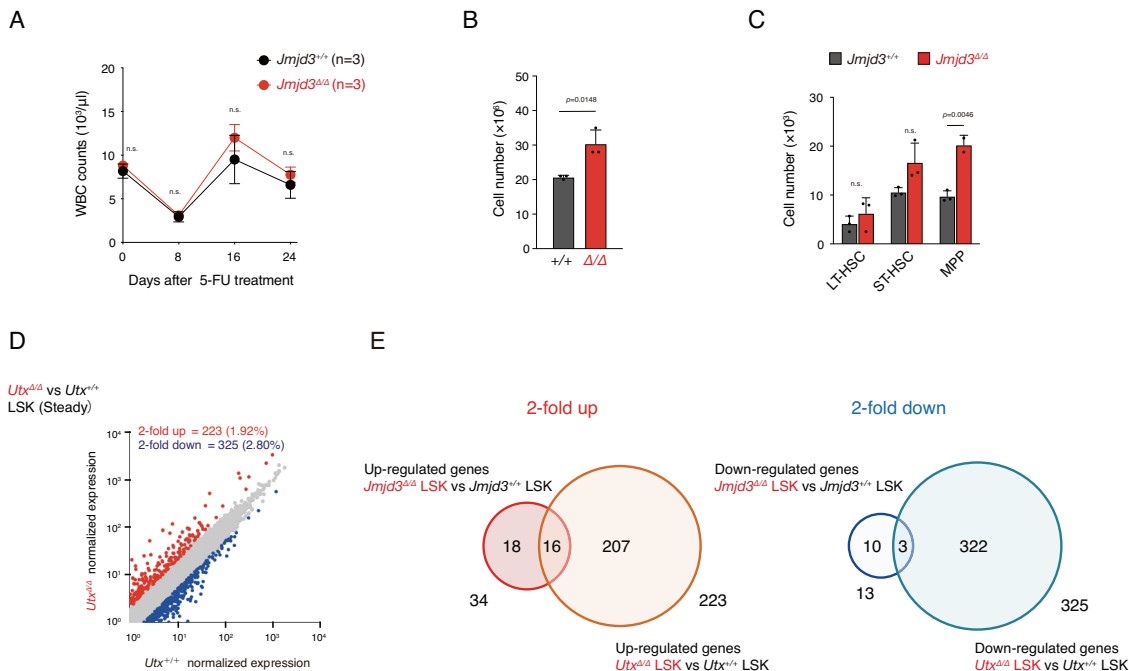

**Figure EV6.** Analysis of *Jmjd3*^Δ/Δ^ HSPCs in replicative stress caused by 5-FU treatment and *Utx*^Δ/Δ^ HSPCs at steady state.

(A) Changes in WBC count in *Jmjd3*^+/+^ and *Jmjd3*^Δ/Δ^ mice every 8 days after 5-FU injection (150 mg/kg) (mean ± SD, n = 3). Student's *t* test was used to calculate *p* value.
(B) Absolute numbers of BM cells from *Jmjd3*^+/+^ and *Jmjd3*^Δ/Δ^ mice 24 days after 5-FU treatment (mean + SD, n = 3). Student's *t* test was used to calculate *p* value. (C) Absolute numbers of HSC subpopulations (LT-HSC, ST-HSC, and MPP) in the BM of *Jmjd3*^+/+^ and *Jmjd3*^Δ/Δ^ mice 24 days after 5-FU treatment (mean + SD, n = 3). Student's *t* test was used to calculate *p* value. (D) Scatter plots comparing normalized expression of individual genes (RPKM > 1) in LSK cells of *Utx*^Δ/Δ^ mice compared with *Utx*^+/+^ mice at steady state. Genes more than twofold upregulated and downregulated are plotted as red and blue dots, respectively. (E) Venn diagrams showing the overlap of genes more than twofold upregulated or downregulated in *Jmjd3*^Δ/Δ^ and *Utx*^Δ/Δ^ LSK cells at steady state.

