## [Peer Review File · EMBO Reports]

JMJD3-mediated senescence is required to overcome stress-induced hematopoietic defects

Hiroaki Honda, Yuichiro Nakata, Takeshi Ueda, Yasuyuki Sera, Miho Koizumi, Katsutoshi Imamura, Akinori Kanai, Ken-ichiro Ikeda, Norimasa Yamasaki, Akiko Nagamachi, Kohei Kobatake, Masataka Taguchi, Yusuke Sotomaru, Tatsuo Ichinohe, Zen-ichiro Honda, Takuro Nakamura, Ichiro Manabe, Toshio Suda, Keiyo Takubo, and Osamu Kaminuma

Corresponding author(s): Hiroaki Honda (honda.hiroaki@twmu.ac.jp) , Yuichiro Nakata (nakatay@chiba-u.jp)

Review Timeline:

Submission Date:	16th Nov 23
Editorial Decision:	15th Jan 24
Revision Received:	11th May 25
Editorial Decision:	27th May 25
Revision Received:	31st May 25
Accepted:	5th Jun 25

Editor: Achim Breiling

Transaction Report:

Dear Prof. Honda,

Thank you for the submission of your research manuscript to EMBO reports. I have now received the reports from the three referees that were asked to evaluate your study, which can be found at the end of this email.

As you will see, the referees think that the findings are of interest. However, they have several comments, concerns, and suggestions, indicating that a major revision of the manuscript is necessary to allow publication of the study in EMBO reports. As the reports are below, and all the referee concerns need to be addressed, I will not detail them here, but, as is apparent from the reports, a better validation of the identified mRNAs and interactions seems necessary to consider the manuscript for publication in EMBO reports.

Given the constructive referee comments, I would like to invite you to revise your manuscript with the understanding that all referee concerns must be addressed in the revised manuscript or in a detailed point-by-point response. Acceptance of your manuscript will depend on a positive outcome of a second round of review. It is EMBO reports policy to allow a single round of revision only and acceptance of the manuscript will therefore depend on the completeness of your responses included in the next, final version of the manuscript.

- 1) a .docx formatted version of the final manuscript text (including legends for main figures, EV figures and tables), but without the figures included. Figure legends should be compiled at the end of the manuscript text.
- 2) individual production quality figure files as .eps, .tif, .jpg (one file per figure), of main figures (up to 8) and EV figures. Please upload these as separate, individual files upon re-submission.

- 4) a complete author checklist, which you can download from our author guidelines

(<https://www.embopress.org/page/journal/14693178/authorguide>). Please insert page numbers in the checklist to indicate where the requested information can be found in the manuscript. The completed author checklist will also be part of the RPF.

5) that primary datasets produced in this study (e.g. RNA-seq, CHIP-seq, structural and array data) are deposited in an appropriate public database. If no primary datasets have been deposited, please also state this in a dedicated section (e.g. 'No primary datasets have been generated and deposited'), see below.

The accession numbers and database should be listed in a formal "Data Availability" section (placed after Materials & Methods) that follows the model below. This is now mandatory (like the COI statement). Please note that the Data Availability Section is restricted to new primary data that are part of this study. This section is mandatory. As indicated above, if no primary datasets have been deposited, please state this in this section

Data availability

8) Regarding data quantification and statistics, please make sure that the number "n" for how many independent experiments were performed, their nature (biological versus technical replicates), the bars and error bars (e.g. SEM, SD) and the test used to calculate p-values is indicated in the respective figure legends (also for potential EV figures and all those in the final Appendix). Please also check that all the p-values are explained in the legend, and that these fit to those shown in the figure. Please provide statistical testing where applicable. Please avoid the phrase 'independent experiment', but clearly state if these were biological or technical replicates. Please also indicate (e.g. with n.s.) if testing was performed, but the differences are not significant. In case n=2, please show the data as separate datapoints without error bars and statistics. See also: <http://www.embopress.org/page/journal/14693178/authorguide#statisticalanalysis>

9) Please add scale bars of similar style and thickness to all the microscopic images, using clearly visible black or white bars (depending on the background). Please place these in the lower right corner of the images themselves. Please do not write on or near the bars in the image but define the size in the respective figure legend.

10) Please also note our reference format:

12) We now use CRediT to specify the contributions of each author in the journal submission system. CRediT replaces the author contribution section. Please use the free text box to provide more detailed descriptions and do not provide your final manuscript text file with an author contributions section. See also our guide to authors: <https://www.embopress.org/page/journal/14693178/authorguide#authorshipguidelines>

13) We would encourage you to use 'Structured Methods', our new Materials and Methods format. According to this format, the Materials and Methods section should include a Reagents and Tools Table (listing key reagents, experimental models, software and relevant equipment and including their sources and relevant identifiers), uploaded as separate file, followed by a Methods and Protocols section in which we encourage the authors to describe their methods using a step-by-step protocol format with bullet points, to facilitate the adoption of the methodologies across labs. More information on how to adhere to this format as well as downloadable templates (.doc or .xls) for the Reagents and Tools Table can be found in our author guidelines (section 'Structured Methods'):

14) Please order the manuscript sections like this, using these names:

Title page - Abstract - Keywords - Introduction - Results - Discussion - Materials and Methods - Data availability section - Acknowledgements - Disclosure and Competing Interests Statement - References - Figure legends - Expanded View Figure legends

I look forward to seeing a revised version of your manuscript when it is ready. Please let me know if you have questions or comments regarding the revision.

Yours sincerely,

Referee #1:

The manuscript by Honda et al. analyzes the role of the histone demethylase JMJD3 in the function of hematopoietic stem and progenitor cells. Using an Mx-Cre knockout approach in mice, the authors demonstrate that JMJD3 loss impaired HSPCs under stress conditions. Altered activity was associated with reduced expression of p16Ink4a, and certain parameters could be rescued or partially rescued by re-expressing p16Ink4a. The respective mechanism was also analyzed in the context of young vs. aged mice, and a JMJD3 inhibitor was used to analyze the importance of JMJD3 in MLL-AF9-induced leukemogenesis. Although some of the results were predictable based on prior studies of the components, this manuscript contains new and interesting findings that extend prior knowledge on this topic. Recommendations are made below to increase rigor and comprehension.

Specific Comments:

1. Abstract - "suggesting a new link for aging and anti-cancer therapies". This is confusing and does not fit with the data nor the specific text of the Abstract.
2. Page 7 - Steady-state hematopoiesis was apparently normal "during the 1.5-year observational period". Mx-Cre often does not often quantitatively excise target loci, and although controls were presented at an early time after Mx-Cre action, as a function of months thereafter, is the hematopoietic system comprised exclusively of JMJD3 null cells or a composite of wild type and mutant cells - with the wild type cells ensuring normal steady-state hematopoiesis and masking phenotypes?
3. Page 8 - "was higher" and many other places in this manuscript - comprehension would increase if numbers were stated e.g., 3.2-fold, with statistics.
4. "transiently confers a proliferative activity" - I am not sure how this conclusion relates to the experiment, and if I am not understanding this, the text would benefit from re-writing.
5. "slightly more" and "significantly reduced" - same comment as above. Please be more specific.
6. Page 9 - "drastic upregulation", Fig 3A - Was this data subjected to statistical analysis?
7. Page 10 - "associated with PRC2 target genes" - It is unclear how this association was established and what statistics support this statement.
8. Page 11, Fig. 3I - Replicates and statistics are required.

9. "marker of cellular senescence" - incorporate the reference.
10. Fig. 4A - Correct the y axis typos.
11. Page 13 - "partially rescued" and "almost fully" - see above regarding numbers and statistics.
12. Page 14 - P16Ink4a expression induced a large decrease in +/- colonies, and this was not discussed.
13. Fig. 6J, "restored by the introduction" - This does not appear to be significant, based on asterisks.
14. General comment about colors on the line graphs - It was very difficult for me to distinguish some of the colors, and since solid symbols were used in all lines on the small graphs, this created confusion.
15. Page 15 and other experiments - The ChIP analysis that led to conclusions about signals at promoters was not accompanied by controls to demonstrate whether signals are promoter-specific or detected at other sites at the loci.
16. Page 16 - "excessive". This is a tiny change - state the percent or fold change.
17. "significantly delayed" - same comment, and this small delay does not provide confidence that inhibiting JMJD3 would have unique utility as an anti-leukemic agent. Furthermore, I do not believe the authors have referenced, described or provided evidence for the specificity of this inhibitor.

Referee #2:

In their manuscript, Nakata and colleagues describe consequences of JMJD3/KDM6B loss in mouse hematopoietic stem and progenitor cells (HSPCs). JMJD3 is a H3K26 demethylase. Upon inducible whole-body knockout in mice generated for this work, the authors do not find obvious phenotypes up to 1,5 years of age at steady state. However, lack of JMJD3 impairs the long-term repopulating capacity of HSPCs in irradiated recipient mice, whereas the short-term repopulating capacity appears moderately enhanced. Similar results were obtained when testing the consequences of JMJD3 loss in MLL-AF9-driven HSPC malignant transformation. Globally, the authors identify 3768 genomic regions with unique H3K26 methylation in JMJD3 knockout GMPs, many of which overlap with recruitment sites of polycomb group proteins (as suggested by literature).

Performing ChIP-qPCR in JMJD3 proficient cells, the authors validate that the tumor suppressor p16INK4a becomes demethylated in repopulating and MLL-AF9 HSPCs, as compared to steady-state HSPCs. Thus, the respective JMJD3 knockout HSPCs show impaired upregulation of p16INK4a, along with reduced b-galactosidase expression, which is often used as a proxy for senescence. Based on these data, the authors conclude that a JMJD3-p16INK4a axis promotes senescence of HSPCs during aberrant excessive proliferation. Additionally, bulk RNASeq analyses reveals that JMJD3 knockout perturbs senescence-associated and promotes proliferation-associated gene expression programs. Importantly, reconstituting JMJD3 knockout HSPCs with p16INK4a or inactive Cyclin D1 restores HSPC stemness upon excessive proliferation or during MLL-AF9-mediated transformation. The authors also find that JMJD3 enhances the expression of p16INK4a in HSPCs during aging.

Finally, the authors explore whether a commercially available JMJD3 inhibitor, GSKJ4, could prevent MLL-AF9-mediated transformation of HSPCs. GSKJ4 treatment appears to suppress senescence via impairing the induction of p16INK4a in HSPCs, and moderately delays leukemia progression.

Overall, the manuscript is well written and contains an impressive collection of data on the topic. These data are in line and extend published information (i) generated from the hematopoietic compartment of inducible JMJD3 knockout mice, and on (ii) a connection between JMJD3 and cellular senescence/p16INK4a. Thus, the current manuscript is of good quality and enhances the understanding of JMJD3 biology in HSPCs in an incremental manner.

Points:

A fraction of the experiments shown seem to be variations of data available in literature. Although cited, some of these data appear undervalued throughout the manuscript, such as the work by Mallaney et al, 2019 (doi: 10.1038/s41375-019-0462-4).

The authors use whole body irradiation to deplete bone marrow recipient mice from HSPCs. Irradiated mice show an extensive inflammatory response. Thus, it is very difficult to distinguish between inflammation-mediated and non-inflammation-mediated effects on HSPC proliferation. This point is of special interest here because others have associated JMJD3 expression and inflammation. The authors should discuss this point.

It is quite hard to conceptualize senescence in regenerating tissues. In their model system, i.e. HSPCs, the authors link p16INK4a expression and senescence by looking at bulk cells. However, it remains unclear whether an individual HSPC that

upregulate p16INK4a really stops to proliferate. Alternatively, p16INK4a upregulation may only induce a transient and not induce a stable cell cycle arrest. Also, the role of cell death in this context has not been addressed. Although challenging, I believe that addressing this point would enhance the manuscript.

The authors identify 3768 regions with unique H3K26 methylation in JMJD3 knockout GMPs. However, they exclusively focus on p16INK4a. The value of the work for the general public would be enhanced by a more thorough investigation and discussion of these interesting results.

Generally, the authors should add the number of biological replicates per genotype in all in vivo reconstitution experiments.

The authors do not rationalize why exclusively the MLL-AF9 oncogene is used to test the consequences of JMJD3 loss in myeloid transformation. This information should be included in the text.

Due to the moderate effects, the experiments shown in Fig. 8 I and J require an increase in n numbers, in particular for the DMSO controls, to solidify the conclusion. It is also not entirely clear how many mice were used for each group.

Concerning the figure legends, the n numbers for all in vitro and in vivo experiments should always be specified for each individual group, i.e. each genotype, treatment group etc., in the respective part of the figure legend. In addition, for all omics analyses the n numbers for biological/technical replicates should always be specified in the respective part of the figure legend.

Referee #3:

In this manuscript, Nakata and colleagues described senescence in hematopoietic stem/progenitor cells (HSPCs) via JMJD3-p16. The authors used a combination of mouse models and molecular analysis to dissect how the histone demethylase JMJD3 regulates p16 expression in mouse HSPCs during regeneration stress conditions and in a mouse model of leukemia (MLL-AF9). In general, the work provides compelling support for their testing hypothesis leading to new insights into how JMJD3 participates in mouse HSPC/AML biology. In large, the experiments were well done however, some re-structuring of the data is really required.

"Major concerns"

-the authors wanted to show that in MLL-AF9 leukemic initiating/propagating/stem cells, which phenotypically is a granulocyte-monocyte derived progenitor (GMP) population, JMJD3 had/has a similar molecular function in these cells as in normal HSPCs. The data seems to indicate this however, it was/is a far stretch comparison not only by doing the contrast with normal HSPCs but importantly some of the HSPC's supporting molecular mechanisms were derived from the L-GMP, which was/is not the most adequate. One relates to normal HSPCs and another is leukemic derived; phenotypically they are also different, with normal HSPCs being LSK (Lin-Sca1+cKit+) and L-GMP being Lin- Sca1-;ckit+, CD34+, CD16/32+; importantly, L-GMP has a fusion protein (MLL-AF9) in which MLL is also a histone demethylase.

It is important to separate the two stories; one with normal HPCS/regeneration and in aging (first 4-5 figures), followed by the AML model with the drug treatment at the end (2-3 figures).

-on a similar note as above, it is important to have supportive molecular data (Figures 3E, F, G, H and I) derived from HSPCs (LSK) to support the HSPC part rather than L-GMP derived. It seems that the authors are "cherry-picking" what to show.

The authors should focus more on providing a strong molecular mechanism (which is the novelty of the paper) on the role of JMJD3-p16 in HSPCs (60-70% of the paper), then hypothesise that a similar mechanism could be happening in certain AMLs, hence using a model of mouse leukemia (MLL-AF9) where JMJD3-p16 regulation is also important. The latter part should be more concise.

Author Responses to Initial Comments:

We thank the Reviewers for their constructive and insightful comments on our manuscript. We revised our manuscript in accordance with the Reviewers' criticisms and addressed the Reviewers' concerns by performing additional experiments and analyses.

We deeply apologize for the delay in resubmitting the revised version. The main reason for the delay is that since the ChIP-grade anti-JMJD3 antibody (ab85392, Abcam) that we used in the experiments of the initial version was no longer commercially available, to address the concerns raised by the Reviewers, we decided to generate a new mouse strain in which a 3 × FLAG tag was integrated into the 3' terminus of the *Jmjd3* locus (referred to as *Jmjd3-Flag* cKI mice) so that the behavior of endogenous JMJD3 can be analyzed by using an anti-Flag antibody (**Fig. 3I** of the resubmitted manuscript).

The replies to each Reviewer's comments are shown on separate pages and in the resubmitted manuscript, to enable easy tracking of the revisions, all the revised and newly added parts are shown in red.

We hope that the resubmitted version has been satisfactorily improved and achieves the priority and quality appropriate for the publication in *EMBO Reports*.

Referee #1: The manuscript by Honda et al. analyzes the role of the histone demethylase JMJD3 in the function of hematopoietic stem and progenitor cells. Using an Mx-Cre knockout approach in mice, the authors demonstrate that JMJD3 loss impaired HSPCs under stress conditions. Altered activity was associated with reduced expression of p16Ink4a, and certain parameters could be rescued or partially rescued by re-expressing p16Ink4a. The respective mechanism was also analyzed in the context of young vs. aged mice, and a JMJD3 inhibitor was used to analyze the importance of JMJD3 in MLL-AF9-induced leukemogenesis. Although some of the results were predictable based on prior studies of the components, this manuscript contains new and interesting findings that extend prior knowledge on this topic. Recommendations are made below to increase rigor and comprehension.

We thank the Reviewer for the encouraging comments that our manuscript contains novel findings concerning the roles of JMJD3 under stress hematopoiesis. In the resubmitted version, we performed additional experiments and analyzed the data based on your constructive comments.

Specific Comments:

1. Abstract - "suggesting a new link for aging and anti-cancer therapies". This is confusing and does not fit with the data nor the specific text of the Abstract.

We thank the Reviewer for the helpful comment. To avoid confusion, we removed the sentence from the **Abstract** of the resubmitted version.

2. Page 7 - Steady-state hematopoiesis was apparently normal "during the 1.5-year observational period". Mx-Cre often does not often quantitatively excise target loci, and although controls were presented at an early time after Mx-Cre action, as a function of months thereafter, is the hematopoietic system comprised exclusively of JMJD3 null cells or a composite of wild type and mutant cells - with the wild type cells ensuring normal steady-state hematopoiesis and masking phenotypes?

We thank the Reviewer for raising this important point. Our data indicate that *Jmjd3* deficient hematopoietic stem/progenitor cells (HPSCs) are particularly vulnerable under stress conditions. The findings raise the possibility that *Jmjd3* deficient HPSCs may also

be gradually eliminated from the bone marrow (BM) over time due to repeated minor stress exposures throughout life. To address this possibility, we investigated the abundance ratio of *Jmjd3* deficient cells at 6 months after pIpC induced *Jmjd3* deletion by determining the knock-out efficiency in the BM. As shown in **Fig. EV1C** of the resubmitted manuscript, *Jmjd3* deficient cells still represented in a substantial proportion in the BM, indicating that they were not progressively excluded.

3. Page 8 - "was higher" and many other places in this manuscript - comprehension would increase if numbers were stated e.g., 3.2-fold, with statistics.

We thank the Reviewer for the instructive comment. Accordingly, we described the degree of alteration with fold changes in line 1 on page 7 of the resubmitted manuscript.

4. "transiently confers a proliferative activity" - I am not sure how this conclusion relates to the experiment, and if I am not understanding this, the text would benefit from re-writing.

We thank the Reviewer for the instructive comment. As we described in line 31 on page 15 of the Discussion section of the initial version, *Jmjd3^{Δ/Δ}* cells exhibited enhanced proliferative capacity during the early phases following stress conditions, such as BMT and oncogene transduction, due to insufficient activation of senescence-associated reprogramming. However, in the late phases following stress conditions, this proliferative capacity was eventually impaired. To avoid confusion, we amended the sentence as follows:

These results indicate that loss of JMJD3 transiently increases proliferative activity on HSPCs during the early phase (1st BMT) but eventually impairs their long-term repopulating and colony-forming abilities in the late phases (2nd to 3rd BMT).

This sentence is included from line 10 to line 13 on page 7 of the resubmitted manuscript.

5. "slightly more" and "significantly reduced" - same comment as above. Please be more specific.

We thank the Reviewer for the insightful comment. Similar to the response to the

comment, we described the degree of alteration with fold changes in lines 21 and 23 on page 7 of the resubmitted manuscript.

6. Page 9 - "drastic upregulation", Fig 3A - Was this data subjected to statistical analysis?

We thank the Reviewer for the helpful comment. The Figure pointed out by the Reviewer shows quantitative real time PCR data, where experimental triplicates were performed, and SD was calculated from technical triplicates. To address this issue, we clarified the use of triplicates in the **Statistical analysis** section in **Materials and Methods** in the resubmitted manuscript.

7. Page 10 - "associated with PRC2 target genes" - It is unclear how this association was established and what statistics support this statement.

We thank the Reviewer for the valuable comment. We identified unique H3K27me3 peaks in *Jmjd3^{Δ/Δ}* cells from whole H3K27me3 peaks detected in both *Jmjd^{+/+}* and *Jmjd3^{Δ/Δ}* cells by CUT&RUN analysis. These unique peaks were then annotated to their nearest genes with the Homer. Enrichr, described in the **Reagents and Tools Table**, was used to understand biological insights, such as functional annotation, pathway enrichment and Transcription Factor Prediction, with the gene lists annotated from the unique peaks. We performed enrichment analysis with ENCODE/ChEA Consensus datasets, including transcriptional and epigenetic factor target information, to rank statistically significant factors associated with the gene lists annotated from the unique H3K27me3 peaks in *Jmjd3^{Δ/Δ}* cells (**Fig. 3F, lower panel**). The results of this analysis indicated that these genes are commonly regulated by JMJD3 and PRC2 (SUZ12 and EZH2).

8. Page 11, Fig. 3I - Replicates and statistics are required.

We thank the Reviewer for the helpful comment. Unfortunately, the ChIP-grade JMJD3 antibody (ab85392, Abcam) that we used in the experiments of the initial version is no longer commercially available. Thus, to overcome this issue, we generated a new mouse strain in which a 3 × FLAG tag was integrated into the 3' terminus of the endogenous *Jmjd3* locus (referred to as the *Jmjd3-Flag* cKI mice) (**Fig. 3I**). This strategy enabled the detection of endogenous JMJD3 protein using an anti-FLAG antibody (**Fig. 3J**).

According to the suggestion from the Reviewer, we performed and replicated ChIP-qPCR analysis using leukemic cells with both anti-FLAG and anti-BMI1 antibodies (**Fig. 3K**). The data indicated that during oncogenic stress from day 1 to day 4, JMJD3 was recruited to the *Cdkn2a* locus where it competed with BMI1 for binding.

9. "marker of cellular senescence" - incorporate the reference.

We thank the Reviewer for the helpful comment. References that demonstrated β -galactosidase as a well-established marker of cellular senescence are cited in the resubmitted manuscript (Dimri *et al*, 1995 and Cai *et al*, 2020).

10. Fig. 4A - Correct the y axis typos.

We apologize to the Reviewer for the mistake. We amended the typos 'galactsidase' to 'galactosidase' in the resubmitted manuscript.

11. Page 13 - "partially rescued" and "almost fully" - see above regarding numbers and statistics.

We thank the Reviewer for the instructive comment. To avoid the confusion pointed out by the Reviewer, we amended sentences which express the degree of alteration with fold changes in line 7 on page 12 of the resubmitted manuscript.

12. Page 14 - P16Ink4a expression induced a large decrease in +/+ colonies, and this was not discussed.

We thank the Reviewer for the valuable comment. Our data indicate that supplementation of p16^{INK4A} in *Jmjd3*^{+/+} HPSCs impairs stem cell potential under stress conditions. It suggests that these cells retain stem cell potential via cellular senescence under stress conditions. To emphasize this consideration, we added the sentence as follows:

Our study indicates that exogenous addition of p16^{INK4A} successfully restores impaired stem cell activity in *Jmjd3*^{Δ/Δ} HSCs and LSCs but reduces stem cell potential in *Jmjd3*^{+/+} HSCs and LSCs (**Fig. 6**). Since *Jmjd3*^{+/+} cells retain stem cell activity via cellular senescence, we postulate that excessive expression of p16^{INK4A} may confer

detrimental effects on stem cell potential under stress conditions.

This sentence is described from line 25 to line 29 on page 16 of the resubmitted manuscript.

13. Fig. 6J, "restored by the introduction" - This does not appear to be significant, based on asterisks.

We thank the Reviewer for the insightful comment. The statistical significance of differences among multiple treatment groups in the Figure was assessed using one-way ANOVA followed by Dunnett's test, as described in **Statistical analysis** of the **Methods** section. When comparing *Jmjd3*^{+/+} cells transduced with *empty* (control) to *Jmjd3*^{Δ/Δ} cells transduced with *p16*^{Ink4a}, the absence of an asterisk indicates that the colony numbers in these groups are comparable (shown in **Fig. 6L** of the resubmitted manuscript, formerly **Fig. 6J** of the initial version).

14. General comment about colors on the line graphs - It was very difficult for me to distinguish some of the colors, and since solid symbols were used in all lines on the small graphs, this created confusion.

We thank the Reviewer for the helpful comment. To avoid confusion, we amended the color of the asterisks on the line graphs pointed out by the Reviewer to be more easily distinguished (**Figs. 6E, 6F, 6H, 6I, 6K, 6L, EV4B, and EV4D** of the resubmitted manuscript).

15. Page 15 and other experiments - The ChIP analysis that led to conclusions about signals at promoters was not accompanied by controls to demonstrate whether signals are promoter-specific or detected at other sites at the loci.

We thank the Reviewer for the constructive comment. H3K27me3 peaks are distributed across various genomic regions including promoters, gene bodies, and intergenic regions. In **Fig. 3G** of the resubmitted manuscript, H3K27me3 peaks were observed in both promoter and gene body regions of the *Cdkn2a* gene. Based on this data, we designed primers to detect H3K27me3 enrichment at this locus. Furthermore, as demonstrated in the ChIP-qPCR analysis in **Fig. 3D**, the H3K27me3 signal at the *Cdkn2a* promoter was substantially higher than at the *p21* promoter, which served as a

negative control, as described in **ChIP-qPCR assay** of the **Methods** section.

16. Page 16 - "excessive". This is a tiny change - state the percent or fold change.

We thank the Reviewer for the helpful comment. To avoid confusion, we amended the sentence pointed out by the Reviewer as follows:

In addition, the population of BrdU⁺ cycling cells exhibited a 1.3 fold increase in GSK-J4 treated L-GMPs compared with DMSO treated L-GMPs (**Fig. EV5H**).

This sentence is described from line 20 to line 22 on page 14 of the resubmitted manuscript.

17. "significantly delayed" - same comment, and this small delay does not provide confidence that inhibiting JMJD3 would have unique utility as an anti-leukemic agent. Furthermore, I do not believe the authors have referenced, described or provided evidence for the specificity of this inhibitor.

We thank the Reviewer for the helpful comment. To provide greater clarity, we modified the sentence as follows:

We performed Kaplan–Meier survival analysis, which revealed a statistically significant difference in the survival periods between the groups.

This sentence is described from line 25 to line 27 on page 14 of the resubmitted manuscript.

In addition, to address the concern pointed out by the Reviewer, additional mice (>5 mice per group) transplanted with leukemic cells were included to increase the sample size. Furthermore, additional references were cited to emphasize the specificity of the drug for JMJD3 inhibition in cancer-related contexts (Ntziachristos *et al*, 2014; Hashizume *et al*, 2014; Lochmann *et al*, 2018). These references are described from line 7 to line 8 on page 14 of the resubmitted manuscript.

Referee #2: In their manuscript, Nakata and colleagues describe consequences of JMJD3/KDM6B loss in mouse hematopoietic stem and progenitor cells (HSPCs). JMJD3 is a H3K27 demethylase. Upon inducible whole-body knockout in mice generated for this work, the authors do not find obvious phenotypes up to 1,5 years of age at steady state. However, lack of JMJD3 impairs the long-term repopulating capacity of HSPCs in irradiated recipient mice, whereas the short-term repopulating capacity appears moderately enhanced. Similar results were obtained when testing the consequences of JMJD3 loss in MLL-AF9-driven HSPC malignant transformation. Globally, the authors identify 3768 genomic regions with unique H3K27 methylation in JMJD3 knockout GMPs, many of which overlap with recruitment sites of polycomb group proteins (as suggested by literature). Performing ChIP-qPCR in JMJD3 proficient cells, the authors validate that the tumor suppressor p16INK4a becomes demethylated in repopulating and MLL-AF9 HSPCs, as compared to steady-state HSPCs. Thus, the respective JMJD3 knockout HSPCs show impaired upregulation of p16INK4a, along with reduced b-galactosidase expression, which is often used as a proxy for senescence. Based on these data, the authors conclude that a JMJD3-p16INK4a axis promotes senescence of HSPCs during aberrant excessive proliferation. Additionally, bulk RNASeq analyses reveals that JMJD3 knockout perturbs senescence-associated and promotes proliferation-associated gene expression programs. Importantly, reconstituting JMJD3 knockout HSPCs with p16INK4a or inactive Cyclin D1 restores HSPC stemness upon excessive proliferation or during MLL-AF9-mediated transformation. The authors also find that JMJD3 enhances the expression of p16INK4a in HSPCs during aging. Finally, the authors explore whether a commercially available JMJD3 inhibitor, GSKJ4, could prevent MLL-AF9-mediated transformation of HSPCs. GSKJ4 treatment appears to suppress senescence via impairing the induction of p16INK4a in HSPCs, and moderately delays leukemia progression. Overall, the manuscript is well written and contains an impressive collection of data on the topic. These data are in line and extend published information (i) generated from the hematopoietic compartment of inducible JMJD3 knockout mice, and on (ii) a connection between JMJD3 and cellular senescence/p16INK4a. Thus, the current manuscript is of good quality and enhances the understanding of JMJD3 biology in HSPCs in an incremental manner.

We thank the Reviewer for the kind understanding and positive evaluation for our manuscript. To further improve the quality of our study, we discussed the novel and key

differences in comparison with a previous report focusing on JMJD3 deletion in the hematopoietic system. Additionally, we explored an alternative hematopoietic stress model, which induces less inflammation than the BMT model based on irradiation.

Points:

A fraction of the experiments shown seem to be variations of data available in literature. Although cited, some of these data appear undervalued throughout the manuscript, such as the work by Mallaney *et al*, 2019 (doi: 10.1038/s41375-019-0462-4).

We thank the Reviewer for the insightful comment. As the Reviewer mentioned, we carefully examined and discussed the novelty of our findings in comparison with the previous study (Mallaney *et al*, 2019), as detailed below:

Mallaney and colleagues reported that *Jmjd3* deficient mice generated using the Vav-Cre system led to defects on HSC repopulation capacity (Mallaney *et al*, 2019), similar to our conditional knockout model using the Mx-Cre system. However, their mouse model exhibited reduced HSCs at steady state that was not observed in our study. The reason for the discrepancy is not clear, but one possibility is the timing of loss of JMJD3, namely, fetal hematopoiesis in that study and adult hematopoiesis in this study. Previous reports also demonstrated that the inherited and acquired loss of the target gene displayed different phenotypes (Rathinam *et al*, 2010 and Nakata *et al*, 2017). Additionally, their model also presented a proliferative advantage of HSCs under inflammatory stress, which was mediated by AP-1 (FOS and JUN) activation. Given that changes in H3K27me3 levels at the *c-Fos* and *c-Jun* loci upon JMJD3 deletion in HSCs were not observed, these findings suggest that the JMJD3-p16^{INK4a} axis may regulate HSC repopulating capacity through upstream cellular senescence *via* H3K27me3 modification, thereby inhibiting AP-1 activation.

This sentence is described from line 28 on page 18 to line 9 on page 19 of the resubmitted manuscript.

The authors use whole body irradiation to deplete bone marrow recipient mice from HSPCs. Irradiated mice show an extensive inflammatory response. Thus, it is very difficult to distinguish between inflammation-mediated and non-inflammation-mediated effects on HSPC proliferation. This point is of special interest here because others have associated JMJD3 expression and inflammation. The authors should discuss this point.

We thank the Reviewer for the insightful comment and we agree that this distinction is challenging to demonstrate. To this end, we performed additional studies using 5-fluorouracil (5-FU)-mediated myelosuppression (Takubo *et al*, 2023), an alternative hematopoietic stress model with relatively lower inflammatory effects compared with irradiation. As shown in **Fig. EV6**, excessive proliferation was observed following 5-FU treatment, resembling the response seen after the 1st BMT with LSK cells (**Fig. 1B**).

It is quite hard to conceptualize senescence in regenerating tissues. In their model system, i.e. HSPCs, the authors link p16INK4a expression and senescence by looking at bulk cells. However, it remains unclear whether an individual HSPC that upregulate p16INK4a really stops to proliferate. Alternatively, p16INK4a upregulation may only induce a transient and not induce a stable cell cycle arrest. Also, the role of cell death in this context has not been addressed. Although challenging, I believe that addressing this point would enhance the manuscript.

We thank the Reviewer for the constructive comment. As initially hypothesized, we examined whether uptake of p16^{INK4a} following treatment with p16-TAT peptide could induce cell cycle arrest under culture conditions. As shown in **Fig. 6C** of the resubmitted manuscript, the proportion of BrdU^{high} HSPCs, which represent actively cycling cells, was significantly reduced following treatment with p16-TAT, suggesting that p16^{INK4A} uptake transiently suppressed cell cycle progression in HSPCs. We hypothesize that this transient cell cycle arrest by cellular senescence, rather than a stable cell cycle arrest, is essential for preserving stemness and overcoming hematopoietic defects and loss of self-renewing and reconstituting activities under stress conditions. Consistently, we observed that stable cell cycle arrest by overexpression of p16^{Ink4a} with retroviral transfection in HSPCs, but not in LSCs, failed to support repopulation (data not shown).

In response to the second concern pointed out by the Reviewer, we addressed whether the loss of p16^{INK4A}-mediated cellular senescence influences apoptosis. As shown in **Fig. 4C**, there was no apparent correlation between the loss of p16^{INK4A}-mediated cellular senescence and cell apoptosis in HSPCs.

The authors identify 3768 regions with unique H3K27 methylation in JMJD3 knockout

GMPs. However, they exclusively focus on p16INK4a. The value of the work for the general public would be enhanced by a more thorough investigation and discussion of these interesting results.

We thank the Reviewer for the valuable comment. To further support the functional role of JMJD3-mediated histone demethylation under stress conditions, we also performed a CUT&RUN assay in BMT-stressed HSPCs. This analysis identified 19,963 unique H3K27me3 peaks in *Jmjd3* deficient HSPCs. Notably, a peak at the *Cdkn2a* locus was found among the unique peaks of both L-GMP (3,768 peaks) and BMT-stressed HSPCs (19,963 peaks). According to the Reviewer's comment, we indicated the presence of a H3K27me3 peak at *Cdkn2a* locus in the Venn diagrams (**Fig. 3F**) and visualized the signal intensity of H3K27me3 peaks at *Cdkn2a* locus with a screenshot of the genome browser (**Fig. 3G**).

Generally, the authors should add the number of biological replicates per genotype in all *in vivo* reconstitution experiments.

We thank the Reviewer for the helpful comment. To avoid confusion, we included the following sentence:

In all experiments, virtually identical sample sizes per group were adjusted to avoid experimental bias.

This sentence is described in the **Statistical analysis** section of **Materials and Methods** in the resubmitted manuscript.

The authors do not rationalize why exclusively the MLL-AF9 oncogene is used to test the consequences of JMJD3 loss in myeloid transformation. This information should be included in the text.

We thank the Reviewer for the instructive comment. To investigate the functions of JMJD3 in HSPCs under oncogenic stress, we utilized the *MLL-AF9* oncogene, as it is a well-established model that accurately defines the leukemic stem cell fraction (Krivtsov *et al*, 2006). Thus, we cited the relevant literature in the resubmitted manuscript. In addition, according to the comment, noting that Reviewer #3 raised a similar concern, we recognized that leukemic stem cells (LSCs) induced by MLL-AF9 arise from the GMP fraction. Therefore, to determine whether JMJD3 deletion similarly affects LSCs originating from a different hematopoietic compartment, we performed a comparative

analysis using a distinct oncogene model. Specifically, we performed RNA-seq analysis on LSCs derived from the LSK fraction driven by *BCR-ABL*, a well-known oncogene which induces chronic myeloid leukemia (CML). Gene sets associated with quiescence and senescence were negatively regulated upon JMJD3 deletion, comparable to those in MLL-AF9-driven LSCs. However, the changes were not statistically significant, as shown in the GSEA plots provided below. These findings suggest a potential oncogene-specific or leukemic type-specific dependency. One possibility is that CML stem cells may possess decreased self-renewal potential compared with normal counterparts (Gishizky *et al.*, 1993).

Schemionek, M., Elling, C., Steidl, U., Bäumer, N., Hamilton, A., Spieker, T., Göthert, J. R., Stehling, M., Wagers, A., Huettner, C. S., *et al.* (2010). BCR-ABL enhances differentiation of long-term repopulating hematopoietic stem cells. *Blood*. 115(16):3185-3195.

Liu, C., Lin, X., Jin, Y., and Pan, J. Protocol for isolation and analysis of the leukemia stem cells in BCR-ABL-driven chronic myelogenous leukemia mice. (2023). *STAR Protoc.* 4(1):102123.

Gishizky, M. L., Johnson-White, J., and Witte, O. N. Efficient transplantation of BCR-ABL-induced chronic myelogenous leukemia-like syndrome in mice. (1993). *Proc Natl Acad Sci U S A.* 90(8):3755-3759.

Due to the moderate effects, the experiments shown in Fig. 8 I and J require an increase in n numbers, in particular for the DMSO controls, to solidify the conclusion. It is also not entirely clear how many mice were used for each group.

We thank the Reviewer for the valuable comment. To address this comment, we

increased sample size to enable a more accurate evaluation, as shown in **Figs. EV5I** and **EV5J** of the resubmitted manuscript.

Concerning the figure legends, the n numbers for all *in vitro* and *in vivo* experiments should always be specified for each individual group, i.e. each genotype, treatment group etc., in the respective part of the figure legend. In addition, for all omics analyses the n numbers for biological/technical replicates should always be specified in the respective part of the figure legend.

We thank the Reviewer for the helpful comment. To address this comment, we specified the sample sizes for each experimental group in the **Figure legends** when they differed between experiments. In addition, we included the sample sizes for the line graphs and survival curves of *in vivo* experiments in the **Figures** where applicable. Regarding the comment for biological/technical replicates, we clarified the use of technical triplicates in the **Statistical analysis** section in **Materials and Methods** in the resubmitted manuscript.

Referee #3: In this manuscript, Nakata and colleagues described senescence in hematopoietic stem/progenitor cells (HSPCs) via JMJD3-p16. The authors used a combination of mouse models and molecular analysis to dissect how the histone demethylase JMJD3 regulates p16 expression in mouse HSPCs during regeneration stress conditions and in a mouse model of leukemia (MLL-AF9). In general, the work provides compelling support for their testing hypothesis leading to new insights into how JMJD3 participates in mouse HSPC/AML biology. In large, the experiments were well done however, some re-structuring of the data is really required.

We thank the Reviewer for finding that our study demonstrates importance of JMJD3-p16^{INK4A} axis on cellular senescence in HSPCs. According to the Reviewers' suggestions, we re-configured the representation of our data the resubmitted manuscript.

"Major concerns"

-the authors wanted to show that in MLL-AF9 leukemic initiating/propagating/stem cells, which phenotypically is a granulocyte-monocyte derived progenitor (GMP) population, JMJD3 had/has a similar molecular function in these cells as in normal HSPCs. The data seems to indicate this however, it was/is a far stretch comparison not only by doing the contrast with normal HSPCs but importantly some of the HSPCs supporting molecular mechanisms were derived from the L-GMP, which was/is not the most adequate. One relates to normal HSPCs and another is leukemic derived; phenotypically they are also different, with normal HSPCs being LSK (Lin-Sca1+cKit+) and L-GMP being Lin- Sca1-;ckit+, CD34+, CD16/32+; importantly, L-GMP has a fusion protein (MLL-AF9) in which MLL is also a histone demethylase.

We thank the Reviewer for the insightful comment. L-GMPs, leukemic stem cells (LSCs), from MLL-AF9-induced AML were derived from the GMP fraction (CD34⁺ CD16/32⁺ c-kit⁺ Sca-1⁻ Lineage⁻) as shown in **Table EV1**. This fraction is distinct from the LSK (c-kit⁺ Sca-1⁻ Lineage⁻) fraction, which is referred to as HSPCs. To demonstrate whether *Jmjd3*-deficient LSCs originating from the LSK fraction exhibit similar defects in stemness and senescence activation as those originating from the GMP fraction, we performed RNA-seq with BCR-ABL-transduced LSCs, which originate from the LSK fraction and give rise to chronic myeloid leukemia (CML) (Schemionek *et al*, 2010 and Liu *et al*, 2023). As shown in the GSEA plots below, gene sets associated with quiescence and senescence from BCR-ABL-driven LSCs were

negatively down-regulated by JMJD3 deletion, similar to those from MLL-AF9-driven LSCs, although the changes were not statistically significant. These findings suggest a potential oncogene-specific or leukemic type-specific dependency. One possibility is that CML stem cells may possess decreased self-renewal potential compared with normal counterparts (Gishizky *et al*, 1993).

Schemionek, M., Elling, C., Steidl, U., Bäumer, N., Hamilton, A., Spieker, T., Göthert, J. R., Stehling, M., Wagers, A., Huettner, C. S., *et al.* (2010). BCR-ABL enhances differentiation of long-term repopulating hematopoietic stem cells. *Blood*. 115(16):3185-3195.

Liu, C., Lin, X., Jin, Y., and Pan, J. Protocol for isolation and analysis of the leukemia stem cells in BCR-ABL-driven chronic myelogenous leukemia mice. (2023). *STAR Protoc.* 4(1):102123.

Gishizky, M. L., Johnson-White, J., and Witte, O. N. Efficient transplantation of BCR-ABL-induced chronic myelogenous leukemia-like syndrome in mice. (1993). *Proc Natl Acad Sci U S A.* 90(8):3755-3759.

It is important to separate the two stories; one with normal HPCS/regeneration and in aging (first 4-5 figures), followed by the AML model with the drug treatment at the end (2-3 figures).

In this manuscript, we aimed to compare HSPCs at steady state with those exposed to BMT-induced and oncogenic-induced stress in parallel as a way to demonstrate the significance of the JMJD3-p16^{INK4A} axis under various stress conditions. To reduce the dependency to the data from the AML model, we transferred the JMJD3 inhibitor data in AML from the main text **Fig. 8** to **Fig. EV5** of the resubmitted manuscript, as

suggested.

-on a similar note as above, it is important to have supportive molecular data (Figures 3E, F, G, H and I) derived from HSPCs (LSK) to support the HSPC part rather than L-GMP derived. It seems that the authors are "cherry-picking" what to show.

We thank the Reviewer for the valuable comment. To address this comment, we performed CUT&RUN assay and β -galactosidase staining with BMT-stressed HSPCs to demonstrate an unbiased approach. As shown in **Figs. 3E, 3F and 3G** as well as **Fig. 4A**, the data from BMT-stressed HSPCs closely resemble those from oncogene-stressed HSPCs.

The authors should focus more on providing a strong molecular mechanism (which is the novelty of the paper) on the role of JMJD3-p16 in HSPCs (60-70% of the paper), then hypothesise that a similar mechanism could be happening in certain AMLs, hence using a model of mouse leukemia (MLL-AF9) where JMJD3-p16 regulation is also important. The latter part should be more concise.

We thank the Reviewer for the instructive comment. To better emphasize the importance of JMJD3-p16^{INK4A}-based mechanisms in HSPCs, we transferred the JMJD3 inhibitor data in AML from the main text **Fig. 8** to **Fig. EV5** of the resubmitted manuscript as suggested.

Dear Prof. Honda,

Thank you for the submission of your revised manuscript to our editorial offices. I have now received the reports from the two referees that were asked to re-evaluate the study, you will find below. As you will see, the two referees now support the publication of your manuscript in EMBO reports. Original referee #1 was unresponsive to my invitation to re-assess the manuscript. However, going through your p-b-p-response, I consider his/her points as adequately addressed.

I have these editorial requests:

- Please remove the running title from the manuscript title page.
 - Please provide the abstract written in present tense throughout.
 - We now use CRediT to specify the contributions of each author in the journal submission system. CRediT replaces the author contribution section. Please use the free text box to provide more detailed descriptions and do NOT provide your final manuscript text file with an author contributions section. See also our guide to authors: <https://www.embopress.org/page/journal/14693178/authorguide#authorshippinguidelines>
 - Then please order the sections like this, using these names:
Title page - Abstract - Keywords - Introduction - Results - Discussion - Methods - Data availability section - Acknowledgements - Disclosure and Competing Interests Statement - References - Figure legends - Expanded View Figure legends
 - Please provide individual production quality figure files as .eps, .tif, .jpg (one file per figure), also of the EV figures. Please upload these as separate, individual files upon re-submission. Please combine some of the figures to have 5-6 final EV figures. Please also update any callouts after these changes.
 - Please check again that the number "n" for how many independent experiments were performed, their nature (biological versus technical replicates), the bars and error bars (e.g. SEM, SD) and the test used to calculate p-values is indicated in the respective figure legends. Please also check that all the p-values are explained in the legend, and that these fit to those shown in the figure. Please provide statistical testing where applicable. Please avoid the phrase 'independent experiment' but clearly state if these were biological or technical replicates. Please also indicate (e.g. with n.s.) if testing was performed, but the differences are not significant. In case n=2, please show the data as separate datapoints without error bars and statistics. See also: <http://www.embopress.org/page/journal/14693178/authorguide#statisticalanalysis>
- If n<5, please show single datapoints for diagrams. Presently several diagrams have no or only partial statistics and miss the 'n.s.'. Please check. Moreover:
- Please note that the exact p values are not provided in the legends of figures 1B-D; 2B, D, E, F, G; 4A-C; 6E, F, H, I, K, L, M; 7D, EV2 D, EV4B-D; EV5 H-J; EV6 B, C
 - Please indicate the statistical test used for data analysis in the legends of figures 1B-D; 2B, D, E, F, G; 3E, 4A-C; 6E, F, H, I, K, L, M; 7D, EV2 D, EV4B-D; EV5 H-J; EV6 B, C
 - Please note that in figures 7D there is a mismatch between the annotated p values in the figure legend and the annotated p values in the figure file that should be corrected.
 - Please add to each legend (main and EV figures, where applicable) a 'Data Information' section (or name the provided section like this) explaining the statistics used or providing information regarding replicates and scales. See: <https://www.embopress.org/page/journal/14693178/authorguide#figureformat>
 - Please add scale bars of similar style and thickness to microscopic images (main and EV figures), using clearly visible black or white bars (depending on the background). Please place these in the lower right corner of the images themselves. Please do not write on or near the bars in the image but define the size in the respective figure legend. Presently, some scale bars seem missing and all have text nearby. Please check.
 - Please upload the Reagents & Tools table as separate file and remove it from the manuscript text. Please also add callouts to the table where appropriate.
 - Please move the primer information (Table EV3) to the Reagents & Tools Table and remove this table from the manuscript files. Please update any callouts.
 - Please upload the other EV tables as separate files.

- Please provide the author checklist with the filled in header info (author name etc. ...).

In addition, I would need from you uploaded separately:

Best,

Referee #2:

The authors have made substantial revisions to the manuscript in response to the reviewers' suggestions. These changes significantly enhance the quality of the work and increase its relevance for the readers of EMBO Reports.

Referee #3:

This revised manuscript contains new and interesting findings that extend prior knowledge on the role of JMJD3 in hematopoiesis. The revised version with extra data and by comparing normal steady HSPCs vs stressed HSPCs (after BMT) and a model of AML seem to read well as a whole.

All editorial and formatting issues were resolved by the authors.

Prof. Hiroaki Honda
Tokyo Women's Medical University
8-1 Kawada-cho, Shinjuku-ku,
Tokyo 162-8666
Japan

Dear Prof. Honda,

I am very pleased to accept your manuscript for publication in the next available issue of EMBO reports. Thank you for your contribution to our journal.

Yours sincerely,
